# Journey to the Centre of Cluster: Harnessing Interior Nodes for A/B Testing under Network Interference

**Qianyi Chen**[1], **Anpeng Wu**[2], **Bo Li**[1],[*] **Lu Deng**[3] **& Yong Wang**[3]
[1]Tsinghua University, [2]Zhejiang University, [3]Tencent Inc.
`{cqy22@mails,libo@sem}.tsinghua.edu.cn`
`anpwu@zju.edu.cn`
`{adamdeng,darwinwang}@tencent.com`

## Abstract

A/B testing on platforms often faces challenges from network interference, where a unit's outcome depends not only on its own treatment but also on the treatments of its network neighbors. To address this, cluster-level randomization has become standard, enabling the use of network-aware estimators. These estimators typically trim the data to retain only a subset of informative units, achieving low bias under suitable conditions but often suffering from high variance. In this paper, we first demonstrate that the interior nodes—units whose neighbors all lie within the same cluster—constitute the vast majority of the post-trimming subpopulation. In light of this, we propose directly averaging over the interior nodes to construct the mean-in-interior (MII) estimator, which circumvents the delicate reweighting required by existing network-aware estimators and substantially reduces variance in classical settings. However, we show that interior nodes are often not representative of the full population, particularly in terms of network-dependent covariates, leading to notable bias. We then augment the MII estimator with a counterfactual predictor trained on the entire network, allowing us to adjust for covariate distribution shifts between the interior nodes and full population. By rearranging the expression, we reveal that our augmented MII estimator embodies an analytical form of the point estimator within prediction-powered inference framework (Angelopoulos et al., 2023a;b). This insight motivates a semi-supervised lens, wherein interior nodes are treated as labeled data subject to selection bias. Extensive and challenging simulation studies[1] demonstrate the outstanding performance of our augmented MII estimator across various settings.

## 1 Introduction

A/B testing has long been the gold standard for modern platforms in deciding whether to launch new product features. However, its basic procedures can easily fail and lead to misleading conclusions when interference exists—specifically when the classic Stable Unit Treatment Value Assumption (SUTVA) is violated, and a unit's potential outcome is influenced by treatments received by adjacent neighbors. Since these influences typically propagate through network topology, such as friendship relations in social networks, we refer to this phenomenon as network interference.

In most industrial A/B testing scenarios, the estimand of interest is the global average treatment effect (GATE), defined as the difference between the mean outcomes under global treatment and global control. Under network interference, we position the estimation of GATE as a nontrivial extrapolation task. Rather than relying on graph-agnostic approaches that drastically reduce dimensionality to one or two dimensions, e.g., Yu et al. (2022b); Cortez-Rodriguez et al. (2024); Bayati et al. (2024), we argue that leveraging the known graph structure is much more appealing.

---

[*]Corresponding author.
[1]The code is available at `https://github.com/Cqyiiii/AMII-Harnessing-Interior-Nodes-for-Network-Experiments`

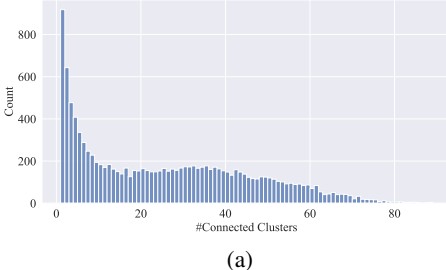
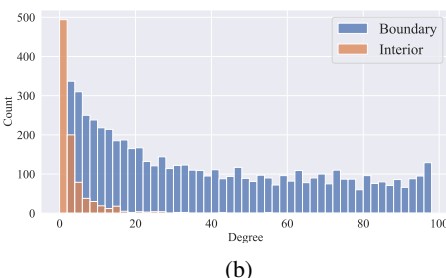

(a)                        (b)

Figure 1: **(a)** Histogram of the number of distinct clusters each unit is connected to. **(b)** Histogram of degree for boundary and interior nodes. The network topology is derived from a Facebook network (Gui et al., 2015), and clusters are generated through Louvain algorithm (Blondel et al., 2008).

As the most basic approach, the difference-in-means estimator is graph-agnostic and thus suffers from severe bias in the presence of interference (Karwa & Airoldi, 2018). This issue is especially pronounced in early-stage experiments, where the treatment proportion typically falls below 10%. In such settings, treated units often have a high proportion of neighbors in the control group, meaning that the treatment vector in their 1-hop ego network can differ significantly from the global treatment.

To leverage the graph topology, the cornerstone methodology is graph cluster randomization (Hudgens & Halloran, 2008; Ugander et al., 2013), which ensures that all units within a cluster receive the same treatment. These clusters are typically pre-generated using graph clustering algorithms. Unlike classic unit-level randomization, graph cluster randomization introduces strong correlations among densely connected units. This allows interior nodes to experience an environment that closely approximates global treatment, thereby facilitating GATE estimation.

Moreover, a widely considered regime is the neighborhood interference assumption (NIA), which informally assumes that interference is restricted to a unit's 1-hop neighbors. Under the NIA and cluster-level randomization, the Horvitz–Thompson (HT) estimator with network exposure indicators (Ugander et al., 2013; Ugander & Yin, 2023) is unbiased. This estimator includes only the outcomes of units that satisfy specific exposure conditions—e.g., having all neighbors receive the same treatment level. When no ambiguity arises, we refer to it simply as the HT estimator and defer its formal definition to Section 2.

Under graph cluster randomization, nodes located in the interior of clusters are always clean and thus are included in the HT estimator. However, these interior nodes represent only a small fraction of the overall population. In contrast, a much larger number of nodes lie along cluster boundaries, yet they are significantly less likely to be selected by the HT estimator. To illustrate this disparity, consider a unit connected to $c$ different clusters, where independent Bernoulli randomization with treatment proportion $p$ is conducted at the cluster level. The probability that this unit is a clean treated node is $p^c$, which can be vanishingly small. To build intuition, Figure 1a displays the distribution of $c$ in our social network. We observe that interior nodes constitute only 8% of the total population. In contrast, boundary nodes often have large values of $c$, leading to extreme inverse probability weights of the form $(1/p)^c$ in the estimator, resulting in highly inflated variance.

There have been numerous efforts to address the issue of explosive variance in the HT estimator. On one hand, refined randomization schemes have been proposed to control the theoretical variance bound of the HT estimator (Ugander & Yin, 2023; Kandiros et al., 2025). However, due to the inherent density of practical social networks, even when sharp theoretical bounds are achieved, the realized variance in practice remains unsatisfactory. On the other hand, alternative estimators have emerged that introduce slight bias in exchange for significant variance reduction, such as the cluster-adaptive estimator (CAE; Liu et al., 2024). CAE also uses exposure indicators to select clean nodes, but replaces inverse probability weighting with a bilevel averaging of outcomes—first within clusters, then across clusters. Nevertheless, we claim that practical clustering algorithms rarely satisfy the strong structural assumptions required by CAE, and that its bilevel averaging step can be further simplified to a single average—leading to substantial variance reduction.

In this paper, we first propose the mean-in-interior estimator, which mimics the difference-in-means estimator by assigning much more moderate weights to unit outcomes, in contrast to the exponential

weights used in the HT estimator—the primary source of its extremely high variance. Our estimator is based on the extensive practice of cluster-level randomization and relies solely on the interior nodes of clusters to compute mean outcomes. Intuitively, these interior nodes tend to reside in a cleaner environment, closer to global treatment or control conditions. With similar assumptions to Liu et al. (2024), specifically that the interior nodes of each cluster are representative enough of the whole cluster, we prove the consistency of the MII estimator.

Furthermore, we observe that the HT, CAE, and MII estimators all exclude most boundary nodes; this exclusion inevitably introduces selection bias when there is a systematic discrepancy between interior and boundary units. To address this challenge, we incorporate a counterfactual predictor trained on the entire network. Leveraging this predictor, we construct an adjustment term for the MII estimator that mitigates covariate mismatch between interior and boundary units. Moreover, by rearranging terms in the estimator, we provide an alternative interpretation through the lens of semi-supervised learning: interior nodes play the role of labeled data—albeit potential selection bias—while boundary nodes contribute representative covariates with partial label information.

**Related Works** Causal inference under interference has been extensively studied in early works (Sobel, 2006; Rosenbaum, 2007). A foundational methodology in this field is graph cluster randomization (Hudgens & Halloran, 2008; Ugander et al., 2013), which allocates treatments at the cluster level and ensures that all units within a cluster receive the same treatment. This approach enhances the statistical performance of several classic estimators, making it a common practice. Additionally, because general interference is often intractable, numerous structural assumptions have been proposed to simplify interference patterns and facilitate analysis.

A significant body of research has centered on the partial interference assumption (Hudgens & Halloran, 2008; Bhattacharya et al., 2020; Forastiere et al., 2021; Candogan et al., 2023), which posits that the network can be partitioned into disjoint clusters, with interference occurring only within each cluster. While this assumption significantly simplifies the interference structure—allowing cluster-level randomization to produce unbiased estimates—it is somewhat restrictive, particularly in settings such as social networks and online marketplaces.

Beyond this special regime, three main structural assumptions have been developed to restrict the form of interference. The first and most widely studied interference pattern in the literature is neighborhood interference (Forastiere et al., 2022; Cortez et al., 2022; Ugander & Yin, 2023; Liu et al., 2024), which assumes that interference is restricted to an individual's direct neighbors. Under this assumption, a unit surrounded by neighbors receiving the same treatment level can be considered as being in a globally treated or control environment, making it well-suited for GATE estimation. With stronger parametric structure, exposure mapping (Aronow & Samii, 2017; Eckles et al., 2016; Ugander et al., 2013; Baird et al., 2018; Vazquez-Bare, 2023) is proposed as a summary of the effect of the entire treatment vector on a specific unit through a predefined function or representation. Furthermore, the potential outcome model is leveraged to develop more elaborate experimental design (Basse & Airoldi, 2018; Yu et al., 2022a; Harshaw et al., 2023; Chen et al., 2023).

Additionally, there is a body of work that leverages graph neural networks (GNNs) as powerful regression models for performing counterfactual prediction on observational data (Leung & Loupos, 2024; Cai et al., 2024; Wu et al., 2025). Specifically, the GATE can be viewed as the difference between two counterfactuals at boundary points, i.e., the global treatment versus the global control.

## 2 BASIC SETTING

### 2.1 PRELIMINARY

We consider a finite population of $n$ units interconnected through an interference network with a known topology. This network is represented as an undirected graph $\mathcal{G} = (\mathcal{V}, \mathcal{E})$, where the node set $\mathcal{V}$ corresponds to the unit set $[n] = \{1, 2, \ldots, n\}$, and the edges are described by the adjacency matrix $A$. We denote the degree of node $i$ by $\deg_i$, and define the $r$-hop neighborhood of unit $i$ as $\mathcal{N}(i, r) = \{j : \ell(i, j) = r\}$, where $\ell(i, j)$ is the shortest path distance between units $i$ and $j$.

We adopt the Neyman–Rubin potential outcomes framework (Rubin, 1974). The treatment assignment is represented by the vector $\mathbf{z} = (z_1, z_2, \ldots, z_n) \in \{0, 1\}^n$. We define the potential outcome for unit $i$ as $Y_i = Y_i(\mathbf{z})$, where $Y_i$ may depend on the treatment assignments of other units.

The parameter of interest is the global average treatment effect (GATE), defined as:

$$\tau := \frac{1}{n} \sum_{i \in [n]} \left( Y_i(\mathbf{1}) - Y_i(\mathbf{0}) \right), \tag{1}$$

where $\mathbf{1}$ and $\mathbf{0}$ denote the $n$-dimensional vectors of ones and zeros, representing the global treatment and global control conditions, respectively.

We now introduce our randomization scheme—cluster-level independent Bernoulli randomization. First, we apply a community detection algorithm to generate clusters, which we take as given in this paper. We denote the resulting clusters as $C_1, C_2, \ldots, C_K$, forming a partition of the node set $\mathcal{V}$.

Second, in each experiment, treatment is assigned at the cluster level, meaning that all units within the same cluster receive the same treatment. The treatment assignments for these clusters are independently drawn from Bernoulli distributions.

Additionally, for a given cluster $C_k$, the interior set is defined as:

$$\text{Int}_k = \{i : i \in C_k, \mathcal{N}(i, 1) \subseteq C_k\}. \tag{2}$$

That is, a node belongs to the interior if all of its 1-hop neighbors are also contained within the same cluster. The boundary set is defined as the complement within the cluster: $\text{Bnd}_k = C_k \setminus \text{Int}_k$. Together, the interior and boundary sets partition the cluster. Intuitively, interior nodes are particularly valuable for estimating the GATE. Under cluster-level randomization, their 1-hop neighbors share the treatment level, making their local environment closely resemble the global treatment or control.

We then formally introduce the neighborhood interference assumption (NIA), which confines interference effect to the 1-hop neighborhood of each unit.

**Assumption 2.1 (NIA)** *We have $Y_i(\mathbf{z}) = Y_i(\mathbf{z}')$ if $\mathbf{z}_j = \mathbf{z}'_j$ holds for all $j \in \mathcal{N}(i, 1)$.*

This assumption is prevalent in the literature, as it imposes a high-level structural constraint on interference while remaining agnostic to the specific form of the exposure function—that is, it does not require a parametric model for how neighboring treatments are aggregated to influence outcomes. Throughout the remainder of this work, we primarily operate under this assumption.

## 2.2 ALTERNATIVE ESTIMATORS

We now introduce three popular alternative estimators against which we will compare our method. The first is the difference-in-means (DIM) estimator, which is interference-agnostic and simply computes the difference in mean outcomes between treated and control units.

$$\hat{\tau}_{DIM} = \frac{z_i Y_i}{\sum_{i \in [n]} z_i} - \frac{(1 - z_i) Y_i}{\sum_{i \in [n]} (1 - z_i)}. \tag{3}$$

The second estimator is the HT estimator with network exposure (Ugander & Yin, 2023).

$$\hat{\tau}_{HT} = \frac{1}{n} \sum_{i \in [n]} \left( \frac{\delta_i(1)}{\mathbb{E}[\delta_i(1)]} - \frac{\delta_i(0)}{\mathbb{E}[\delta_i(0)]} \right) Y_i, \tag{4}$$

where $\delta_i$ is the exposure indicator: $\delta_i(z_0) = \mathbb{I}\{\sum_{j \in \mathcal{N}(i,1)} z_j / \deg_i = z_0, z_i = z_0\}$. This definition implies that the HT estimator relies solely on the outcomes of clean nodes—those whose neighbors all receive the same treatment level as the node itself. This exposure condition is commonly referred to as full-neighborhood exposure (Ugander et al., 2013). Under the NIA, the HT estimator is guaranteed to be unbiased, making it a theoretically appealing choice.

Finally, we introduce a more nuanced estimator, the CAE (Liu et al., 2024). Like the HT estimator discussed earlier, CAE uses the same indicator to select clean nodes. However, instead of applying inverse probability weights, it computes a bilevel average over the observed outcomes. As such, it

can be interpreted as a hybrid of the DIM and HT estimators. To define it, let $t_k$ denote the treatment assigned to cluster $C_k$. We first compute the within-cluster average:

$$\hat{Y}_{k,z_0} = \sum_{i \in C_k} \frac{\delta_i(z_0) Y_i}{\sum_{j \in C_k} \delta_j(z_0)}. \tag{5}$$

Then, an outer average is taken across clusters, yielding the CAE:

$$\hat{\tau}_{CAE} = \frac{\sum_{k \in [K]} t_k Y_{k,1}}{\sum_{l \in [K]} t_l} - \frac{\sum_{k \in [K]} (1 - t_k) Y_{k,0}}{\sum_{l \in [K]} (1 - t_l)}. \tag{6}$$

## 3 HARNESSING THE INTERIOR NODES

### 3.1 MEAN-IN-INTERIOR ESTIMATOR

Now we introduce our MII estimator, which is the difference between the outcomes of treated and control interior nodes. Here, $\mathrm{Int} = \cup_{k \in [K]} \mathrm{Int}_k$ denotes the union of the interior sets across clusters.

$$\hat{\tau}_{MII} = \frac{\sum_{i \in \mathrm{Int}} z_i Y_i}{\sum_{j \in \mathrm{Int}} z_j} - \frac{\sum_{i \in \mathrm{Int}} (1 - z_i) Y_i}{\sum_{j \in \mathrm{Int}} (1 - z_j)}. \tag{7}$$

As an estimator based on the difference-in-means approach, the MII estimator assigns moderate weights to observed outcomes. Empirically, we find that the bias of the CAE estimator is similar to that of the MII estimator, while the MII estimator exhibits substantially lower variance. Furthermore, under assumptions similar to those in Liu et al. (2024), we demonstrate that the MII estimator is consistent, with proof provided in Appendix B.1.

**Assumption 3.1** *Technical assumptions for the consistency of MII estimator.*

1. *The proportion of interior nodes becomes asymptotically uniform across all clusters:*

$$\max_{i \in [K]} \left| \frac{|\mathrm{Int}_i|}{\sum_{k \in [K]} |\mathrm{Int}_k|} - \frac{n_i}{\sum_{k \in [K]} n_k} \right| = o_p \left( \frac{1}{K} \right). \tag{8}$$

2. *The interior nodes provide a sufficiently representative sample of each cluster:*

$$\begin{aligned} \max_{k \in [K]} |\bar{Y}_k(\mathbf{1}) - \bar{Y}_{k,Int}(\mathbf{1})| = o_p(1), \\ \max_{k \in [K]} |\bar{Y}_k(\mathbf{0}) - \bar{Y}_{k,Int}(\mathbf{0})| = o_p(1). \end{aligned} \tag{9}$$

*Here, $\bar{Y}_k(\mathbf{1})$ and $\bar{Y}_{k,\mathrm{Int}}(\mathbf{1})$ denote the mean outcomes within cluster $k$ and within the interior nodes of cluster $k$, respectively. The asymptotics is with regard to the number of clusters, $K$.*

**Theorem 3.1** *Suppose Assumption 2.1 and 3.1 hold; then the MII estimator is consistent, i.e.,*

$$\hat{\tau}_{MII} - \tau = o_p(1). \tag{10}$$

We first note that the two parts of Assumption 3.1 specify uniformity across clusters and the representativeness of interior nodes, respectively. Notably, our requirements are considerably weaker than the corresponding assumptions in Liu et al. (2024)—who, for instance, assume mean outcomes within each cluster to be an identical constant. This discrepancy arises partly because our current focus is limited to consistency, whereas their work aims to establish a Central Limit Theorem (CLT).

In practice, deriving a CLT and performing subsequent statistical inference under network interference is intrinsically difficult for global estimands like GATE (Kandiros et al., 2025). With specific experimental design and estimator, even if one accepts various unverifiable technical assumptions, there remain stringent restrictions on network density; for example, the maximum degree of the two-hop connected graph $\mathcal{G}^2$ should be $o(n^{1/9})$ (Kandiros et al., 2025).

In summary, compared to bias analysis, characterizing variance is typically far more complex due to the intricate interdependencies induced by the network. To provide intuition, we illustrate the variance of the MII estimator using the following potential outcome model:

$$Y_i(\mathbf{z}) = \beta z_i + h\left(\frac{\sum_{j \in \mathcal{N}(i,1)} z_j}{\deg_i}, v_i\right). \tag{11}$$

Here, $\beta$ is the direct treatment effect, covariate $v_i$ is i.i.d. across all units, and $h$ is the interference function. This model incorporates the classic linear-in-means (Aronow & Samii, 2017) and polynomial interference model (Cortez et al., 2022) as special cases.

An important property of this class of potential outcome models is that each interior node provides an unbiased estimate of the mean outcomes under global treatment or control, regardless of the complexity of $h$. This suggests simply averaging over all interior nodes with equal weights. Per the Cauchy-Schwarz inequality, equal weighting yields the minimum-variance unbiased estimator when only interior nodes are available. In principle, adding boundary nodes can further reduce variance. However, under network interference, doing so necessitates delicate weights—e.g., inverse-propensity weights for network exposure in HT estimator or bilevel cluster-relevant weights in CAE—which can actually introduce additional inefficiency.

## 4 ADJUSTING FOR COVARIATE DISTRIBUTIONAL DISCREPANCY

### 4.1 MOTIVATION

To begin, we argue that structural assumptions imposed on clusters—such as the representativeness of interior nodes or a constant expectation for within-cluster mean outcomes (Liu et al., 2024)—are often fragile, precisely because the clustering algorithms used in practice tend to be coarse.

In practice, a scalable algorithm for generating clusters is usually a heuristic one. Graph clustering for minimizing certain objectives is generally a challenging task, often involving NP-hard combinatorial optimization problems, such as correlation clustering (Pouget-Abadie et al., 2019) and balanced clustering (Brennan et al., 2022). In practice, social platforms with hundreds of millions of users typically construct millions of clusters to meet the immense demand for A/B tests, which can number in the thousands weekly. This large-scale problem necessitates heuristic solutions or fast but heuristic algorithms, rendering more elaborate algorithms, such as causal clustering (Viviano et al., 2025), which involves solving semidefinite programming, inapplicable.

Indeed, on a billion-scale social platform, we observe that certain user engagement metrics—among the primary outcome variables recorded in nearly all experiments—exhibit clear discrepancies between interior and boundary subpopulations. Desensitized empirical evidence of this phenomenon is shown in Figure 2. However, as discussed in our introduction, most estimators without a regression component tend to rely heavily on interior nodes, e.g., referring back to Equations (4) and (6).

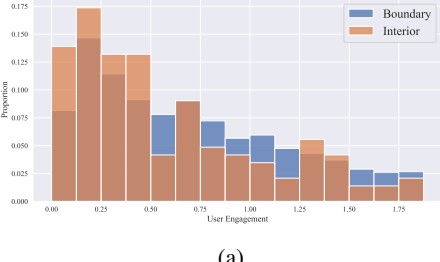
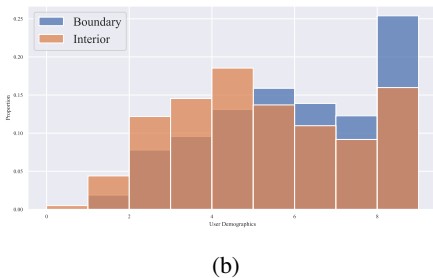

(a)                                                  (b)

Figure 2: Distribution of **(a)** an outcome variable (user engagement) and **(b)** a demographic attribute across interior and boundary units. Data are collected from a billion-scale social platform.

## 4.2 Bridging the gap with counterfactual predictor

Since our MII estimator also employs moderate weighting rather than inverse propensity weighting, it is similarly susceptible to the aforementioned selection bias. To address this, we propose training a counterfactual predictor over the entire graph. Specifically, we introduce the notion of counterfactual prediction: given a treatment vector $\mathbf{z}$, covariate matrix $X$, and network topology $A$, we define a function $f(\mathbf{z}, X, A)$ that predicts outcomes for all units under treatment assignment $\mathbf{z}$. The function $f$ can be flexibly parameterized to fit the data; for example, it may be instantiated as a GNN.

Given the observed outcomes $\mathbf{Y}(\mathbf{z})$ under randomization, we train the function $f$ via regression. We then shift the treatment regime to $\mathbf{z} = \mathbf{1}, \mathbf{0}$, and define the augmented MII estimator as:

$$
\hat{\tau}_{AMII} = \hat{\tau}_{MII} + \left( \frac{1}{n} \sum_{j \in [n]} f(\mathbf{1}, X, A)_j - \frac{1}{s_1} \sum_{i \in \text{Int}} z_i f(\mathbf{1}, X, A)_i \right)
$$
$$
- \left( \frac{1}{n} \sum_{j \in [n]} f(\mathbf{0}, X, A)_j - \frac{1}{s_0} \sum_{i \in \text{Int}} (1 - z_i) f(\mathbf{0}, X, A)_i \right). \tag{12}
$$

Here, $s_1$ ($s_0$) denotes the number of treated (control) units within the interior set. The adjustment term captures the difference between the average predicted outcomes for the full population and those for the treated (control) interior nodes. To illustrate the idea behind the AMII estimator, we consider the following potential outcome model, which includes an additional interaction term between the treatment and a covariate that can be network-dependent.

$$
Y_i(\mathbf{z}) = (\beta + \alpha u_i)z_i + h\left( \frac{\sum_{j \in \mathcal{N}(i,1)} z_j}{\deg_i}, v_i \right). \tag{13}
$$

Here, the $v_i$ are covariates that may be i.i.d., but we allow $u_i$ to be network-dependent and thus non-i.i.d.; for example, $u_i$ could represent the normalized degree $\deg_i / \overline{\deg}$, whose distribution differs between the boundary and interior subpopulations. Additionally, $\beta$ and $\alpha$ are unknown scalar parameters, and $h$ is an unknown interference function, which may be non-linear. Given this model and the form of the AMII estimator, we provide the following intuition: regression models with suitably chosen inputs can often capture the interaction term effectively. However, in early-stage experiments (small $p$), they may struggle to extrapolate the interference function $h$—precisely the component where the MII estimator demonstrates a comparative advantage.

More broadly, we identify the extrapolation of the interference term as the central challenge in estimating the GATE under network interference. In practice, we typically observe outcomes under low treatment proportions (e.g., $p = 0\%, 5\%, 10\%$), while our goal is to extrapolate to a fully treated scenario ($p = 100\%$). A key observation is that for boundary units $i$, the network exposure term $\sum_{j \in \mathcal{N}(i,1)} z_j / \deg_i$ tends to concentrate around $p$, which reflects a regime far from global treatment. This highlights that the main extrapolation challenge lies in recovering the interference function $h$. As also shown in Chen & Li (2024), the performance of naive regression-based estimators can degrade significantly when $h$ is nonlinear (e.g., square root, quadratic), in contrast to the linear case.

Next, we provide a formal characterization of the intuition of AMII estimator, highlighting the bias reduction achieved by the adjustment term. To manage the complexity arising from network dependencies and the functional form of the regression, we impose parametric assumptions on $f$.

**Assumption 4.1 (Regression Model)** *Given $n$ samples $\{(u_i, v_i, z_i, Y_i)\}_{i=1}^n$, we assume the trained regression function $f$ has the form:*

$$
f(\mathbf{z}, X, A)_i = (\hat{\beta}_n + \hat{\alpha}_n u_i)z_i + \text{MEAN}(\{g_\theta(z_j, v_j) \mid j \in \mathcal{N}(i,1)\}). \tag{14}
$$

*Here, $\hat{\beta}_n, \hat{\alpha}_n$ denote the estimated coefficients of the linear part, $g_\theta$ is a learnable transformation function (e.g., a multilayer perceptron), and $\text{MEAN}$ denotes the mean pooling of a set of real values.*

This functional form mimics a partial linear model. We assume the linear part is correctly specified, while the non-linear part—the interference function—may be misspecified. For this non-linear part, we apply a form of one-layer graph convolution.

**Theorem 4.1** *Given Assumption 4.1, potential outcome model in Equation (13), and network topology $\mathcal{G}$, the biases of the MII estimator and the AMII estimator are given by:*

$$\begin{aligned}
\text{Bias}(\hat{\tau}_{MII}) &= \alpha(\mu_{\text{Int}} - \mu) \\
\text{Bias}(\hat{\tau}_{AMII}) &= (\mathbb{E}[\hat{\alpha}_n] - \alpha)(\mu - \mu_{\text{Int}}),
\end{aligned} \tag{15}$$

*where $\mu = \mathbb{E}[u_i]$, $\mu_{\text{Int}} = \mathbb{E}[u_i \mid i \in \text{Int}]$.*

The proof of Theorem 4.1 is provided in Appendix B.2. In summary, we observe that the bias in MII arises from the discrepancy between the interior and the entire population. Furthermore, when the linear component is correctly specified, the quantity $|\mathbb{E}[\hat{\alpha}_n] - \alpha|$ is typically much smaller than $|\alpha|$, indicating a substantial reduction in bias. Additionally, we find that the AMII estimator is a harmless adjustment in the absence of distributional differences between the interior and the full population; that is, it does not introduce any additional bias.

Since these results rely on parametric assumptions, we evaluate the performance of the AMII estimator through a systematic simulation study where structural assumptions are violated in various ways. Despite this, the AMII estimator demonstrates robustness and outstanding performance.

### 4.3 A TALE OF SEMI-SUPERVISION

We then present another interpretation of our method. For simplicity, we focus on estimating the mean outcome under global treatment. By rearranging the expression of AMII estimator, we obtain:

$$\hat{\tau}_{AMII,1} = \frac{1}{n} \sum_{j \in [n]} f(\mathbf{1}, X, A)_j + \frac{1}{s_1} \sum_{i \in \text{Int}} z_i \left( Y_i - f(\mathbf{1}, X, A)_i \right). \tag{16}$$

This embodies a form of the point estimator in PPI (Angelopoulos et al., 2023a;b), and we interpret our estimator within that sound framework. Specifically, we treat the outcomes of treated interior nodes as reliable labeled samples, while the outcomes of boundary nodes are used solely for training the predictor $f$. The goal is to estimate the population mean outcome (i.e., the mean label), using a small subset of labeled data (interior units) alongside a large pool of unlabeled samples, for which only covariates are observed. One of the core ideas behind PPI is to correct the bias of pure model-based predictions using a small set of true labels, thereby mitigating estimation error. This perspective aligns with this reorganized form of our estimator.

However, due to the complexity of network interference, the mean of "true labels" still incurs bias. As noted earlier and illustrated in Figure 2, this bias is sourced from the formation of clusters, i.e., there exists a substantial distributional discrepancy between interior and boundary units, in contrast to the missing-completely-at-random setting in PPI framework. Hence, our AMII estimator should be first positioned as **debiasing using prediction**, by which we tackle such selection bias of mean of "true labels". In comparison, the core idea of both PPI point estimator and the classic doubly-robust estimator is debiasing the mean of predictions using a few labeled samples.

## 5 SIMULATION STUDY

### 5.1 BASIC SETUP

First, since the point of our methodology is the reduction of mean squared error (MSE) rather than purely variance reduction, we must demonstrate the effect of bias reduction, especially for the proposed AMII estimator. However, due to the complex effect in real platform experiments, e.g., existence of many parallel experiments, temporal effect, launch of other new traits, etc., we have no access to the ground truth of GATE, which hinders the evaluation of bias. Therefore, we follow the convention of existing literature and conduct a Monte Carlo simulation with 1,000 repetitions, where randomness arises from treatment allocation (randomization) and exogenous noise in the potential outcomes. We then assess the bias, variance, and MSE based on these repetitions.

**Data.** We use a Facebook social network[2] consisting of 11,586 nodes and 568,309 edges. To generate clusters, we apply the Louvain algorithm and report results using a resolution parameter

---

[2]The network topology is available at https://networkrepository.com/socfb-Stanford3.php.

$\gamma = 5$, which yields 95 clusters. The resolution primarily affects the number of clusters and has a minor effect on the proportion of interior nodes (about 8%). Based on this clustering, we implement independent Bernoulli randomization at the cluster level and evaluate three treatment proportions, $p \in \{0.1, 0.3, 0.5\}$, corresponding to the three-stage experiments conducted prior to the launch.

We focus on the estimation of the mean outcome under global treatment, and set $Y_i(\mathbf{0}) = 0$ for simplicity. When this assumption does not hold, a simple adjustment—subtracting the baseline level (Yu et al., 2022a)—can transform the problem into this setting.

**Potential outcome model.** We adopt the 2-hop interference framework from Chen & Li (2024) and additionally include two covariates: degree and number of distinct clusters each unit is connected to, denoted by $\deg_i$ and $c_i$, respectively. The distributions of these covariates differ between interior and boundary nodes, thereby inducing a violation of the representativeness assumption.

$$\mathbf{Y}(\mathbf{z}) = \beta \mathbf{z} + B\mathbf{z} + \frac{1}{2}(\frac{\deg}{\overline{\deg}} + \frac{\mathbf{c}}{\bar{c}}) \odot \mathbf{z} + \sigma \epsilon. \tag{17}$$

Here, $B$ is the interference matrix, defined as:

$$B = (\mathbf{1}\mathbf{1}^\top - I_n) \odot \left( \sum_{l=1}^{2} r_l (D^{-1}A)^l \right). \tag{18}$$

In this formulation, $D$ is the diagonal degree matrix, and $\odot$ denotes the element-wise product. This model naturally incorporates a linear 2-hop interference, with the intensity characterized by $r_2$. The diagonal element of $B$ is removed to avoid confusion with direct treatment effect.

Moreover, two normalized covariates interact with the treatment vector $\mathbf{z}$. Finally, $\epsilon \sim \mathcal{N}(\mathbf{0}, I_n)$ represents an exogenous Gaussian noise, and $\sigma$ controls the intensity of this noise.

**Parameter setting.** We set $(\beta, r_1) = (1, 1)$, and consider two levels of 2-hop interference, $r_2 \in \{0, 1\}$, corresponding to the case that NIA holds and substantial 2-hop interference exists, respectively. Additionally, following Liu et al. (2024), we examine a setting with a relatively low signal-to-noise ratio by setting $\sigma = 2$. This scenario is particularly relevant in industrial practice, where most new product traits tend to have relatively minor effects on the outcomes of interest.

**Baselines.** We evaluate five methods in total: the Hájek estimator, the cluster-adaptive estimator (CAE), the mean-in-interior (MII) estimator, the counterfactual prediction method using a graph neural network (GNN), and the augmented MII estimator (AMII).

We now provide some additional notes. Since the variance of the HT estimator tends to be explosive in complex scenarios, we adopt its variant—the Hájek estimator (Ugander & Yin, 2023)—which applies self-normalization to the weights, substantially reducing variance at the cost of a small increase in bias. Given that the potential outcome model is fundamentally linear, we use three Chebyshev convolution layers (Defferrard et al., 2016) to construct the GNN, without incorporating any non-linear activation functions. Overall, this is not intended to be a delicate architecture.

## 5.2 RESULTS AND DISCUSSION

Due to space constraints, we present only the results using clustering resolution $\gamma = 5$ in the main paper, as shown in Figure 3. The complete experiments—including alternative **resolution levels**, different **covariate specifications**, different **network topology**, and a series of ablation studies—are reported in Appendix A. Here, we summarize the main conclusions from our results.

First, we observe the outstanding performance of the AMII estimator: it achieves substantially lower bias while maintaining variance comparable to that of the three estimators without a regression component. This translates to a much lower MSE, even when $p = 0.1$ and GNN perform poorly. Notably, the bias reduction becomes even more pronounced in the presence of 2-hop interference (i.e., $r_2 = 1$). Furthermore, as the coefficient of the interaction term increases from 0.5 to 1, the advantage of AMII becomes even more distinct.

Second, we examine the performance of the GNN estimator. A notable pattern is its strong dependence on a relatively high treatment proportion, under which substantially more units become informative for estimating outcomes under global treatment.

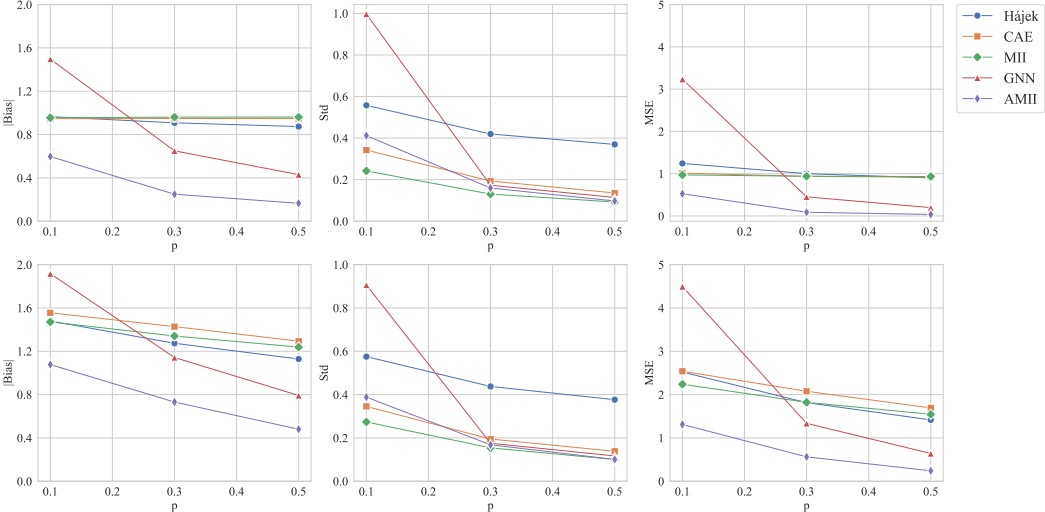

Figure 3: Statistical performance with $r_2 = 0$ (first row) and $r_2 = 1$ (second row).

Next, we investigate the performance of the MII estimator. It consistently achieves the lowest variance across all settings, while exhibiting a similar level of bias as the Hájek estimator and CAE. This bias is expected due to the influence of covariates $\deg_i$ and $c_i$. In the absence of the interaction term and 2-hop interference (see Table 10), we find that MII is nearly unbiased—similar to CAE—while still maintaining the lowest variance, making it the best one there. Nonetheless, the AMII estimator also performs well. This demonstrates a form of harmlessness property of the adjustment term.

Moreover, we discuss the impact of clustering resolution. The corresponding statistics are provided in Table 1, and results are presented in Tables 2 through 7. The behavior of bias is difficult to analyze since the pattern of between-cluster edges changes during the cluster formation process. Regarding variance, we note that the standard deviation of estimators generally decreases as $\gamma$ increases from 2 to 5 (except for the Hájek estimator), as the number of "cluster-level samples" increases. Nonetheless, there exists an irreducible component of variance, as demonstrated by the fact that estimator variances remain almost unchanged when the number of clusters further doubles ($\gamma$ increases from 5 to 10). This differs substantially from classical regimes and again illustrates the complicated variance structure under network interference.

Last, we discuss an additional ablation study that examines whether training $f$ on the entire graph or only on the boundary nodes yields better performance. The latter is a natural sample-splitting idea, but we note that the complex dependency in network data makes it hard to tease out a clear story from this choice. The related results are reported in Tables 12 and 13. Empirically, we find that when the treatment proportion is low (e.g. $p = 0.1$) training with the full population is much better. When $p$ increases, the performance of these two approaches becomes similar. The intuition behind this result is that boundary nodes become significantly more informative as $p$ increases; for instance, the expected exposure $\mathbb{E}[\delta_i(1)] = p^{c_i}$ grows rapidly for nodes with a high level $c_i$. This implies that as $p$ rises, the estimator captures more signal from the local neighborhood structures, effectively leveraging the boundary information that is otherwise sparse in low-exposure regimes.

## 6 CONCLUSION

In this paper, based on extensive practice of cluster-level randomization, we systematically identify the limitations of existing estimators in GATE estimation and propose the mean-in-interior estimator to eliminate unnecessary reweighting, achieving further variance reduction. Recognizing the heavy reliance on the interior nodes and potential selection bias sourced from clustering, we further propose an adjustment term and enhance its position through a novel semi-supervision perspective. We summarize the adjustment as "debiasing using predictions", which is of independent interest for correcting selection bias. Through a series of challenging simulation studies, we demonstrate the remarkable performance of our methodology, especially in the early-stage experiment.

## ACKNOWLEDGMENTS

This research was supported by the National Natural Science Foundation of China (No.72171131).

## ETHICS STATEMENT

This work adheres to the ICLR Code of Ethics. Our study does not involve human subjects, personally identifiable information, or sensitive data. All datasets used are publicly available and widely adopted in the research community, and we followed recommended practices to ensure fairness, privacy, and reproducibility. The methods we propose are intended solely for academic research and are not designed to cause harm or enable misuse. We are not aware of any conflicts of interest, sponsorship concerns, or ethical risks associated with this work.

## REPRODUCIBILITY STATEMENT

We have made extensive efforts to ensure the reproducibility of our results. All code and scripts necessary to reproduce the experiments are provided in the supplementary material. Complete proofs of theoretical results are also included in the appendix. Together, these resources are intended to enable independent verification and facilitate future research building upon our work.

## USAGE OF LARGE LANGUAGE MODEL (LLM)

The authors employed LLMs solely to refine wording and correct grammatical errors in the final manuscript. All ideas, analysis, and conclusions are the authors' own.

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

# A    DETAILED SIMULATION RESULTS

In this section, we provide detailed numerical results. The default setting has been introduced in the main paper; specifically, the clustering resolution is set to $\gamma = 5$, and the potential outcome model follows Equation (17).

## A.1    IMPACT OF CLUSTERING RESOLUTION

We first examine the influence of the clustering resolution $\gamma$, as all five methods rely on cluster-level randomization. Broadly speaking, $\gamma$ controls the modularity gain required for merging during the hierarchical clustering process. A higher value of $\gamma$ corresponds to a lower modularity gain, which tends to preserve more small clusters.

We report the basic statistics for three levels of $\gamma$, including the number of clusters, the proportion of interior nodes, and the proportion of within-cluster edges.

Table 1: The statistics of different clustering schemes.

| $\gamma$ | #clusters | %interior | %within-clusters |
|---|---|---|---|
| **2** | 30 | 8.4 | 41.6 |
| **5** | 95 | 7.9 | 31.7 |
| **10** | 192 | 7.2 | 26.6 |

We then report the results for $\gamma \in \{2, 5, 10\}$ from Table 2 to Table 7. As a reminder, the default setting reported in the main paper is $\gamma = 5$, which represents a moderate level. The main takeaway here is that clustering resolution has only a minor effect on the performance of our methodology, suggesting that the choice of $\gamma$ can be flexibly guided by other considerations.

## A.2    IMPACT OF COVARIATE SPECIFICATION

In this subsection, we examine the impact of different covariate specifications and the weight of the interaction term. Before presenting the detailed results, we clarify that the input features of the GNN consist only of node degree $\deg_i$ and treatment assignment $z_i$. This implies that, under the default setting, the GNN explicitly includes one of the covariates—degree—as part of its input. Accordingly, we first investigate the following potential outcome model, which removes the normalized degree from the potential outcome model:

$$\mathbf{Y}(\mathbf{z}) = \beta \mathbf{z} + B\mathbf{z} + \frac{\mathbf{c}}{\bar{c}} \odot \mathbf{z} + \sigma \epsilon. \tag{19}$$

In this model, the GNN no longer has direct access to the covariate that interacts with the treatment assignment. The parameter setting remains the same as in the main paper, and we report the results in Table 8 and Table 9. Compared to Table 3 and Table 6, we find that the impact of this covariate specification is also minor. This suggests that even when the GNN does not explicitly leverage covariates derived solely from network topology, its performance remains largely unaffected. Nonetheless, we note that demographic features, which cannot be inferred from the network structure, should be provided as input covariates for regression.

Next, we consider varying the weight of the interaction term, which is set to $0.5$ in the default configuration. From the results in Table 3 and Table 6, we observe that the CAE, Hájek, and MII estimators all struggle to effectively address covariate distribution discrepancies, even as the treatment proportion increases. This suggests that, as the interaction weight increases, the advantage of the AMII estimator becomes more pronounced. Therefore, we omit experiments with higher weights and instead turn to the opposite scenario—zero weight—where the interaction term is absent.

The potential outcome model used in this setting is as follows, and the corresponding results are reported in Tables 10 and 11.

$$\mathbf{Y}(\mathbf{z}) = \beta \mathbf{z} + B\mathbf{z} + \sigma \epsilon. \tag{20}$$

When NIA holds (i.e., $r_2 = 0$), we observe that the CAE, Hájek and MII estimators all achieve near-unbiasedness, consistent with theoretical expectations. In this setting, the MII estimator attains

substantially lower variance, making it the best-performing method in this case. In contrast, the GNN estimator does not benefit significantly from the simplicity of this setting—it still requires a larger volume of high-quality data (or a higher treatment proportion) to perform well. Interestingly, despite the poor performance of the GNN estimator, the AMII estimator achieves results comparable to MII. This phenomenon demonstrates a form of harmlessness property, further validating the robustness of our methodology.

In the scenario where $r_2 = 1$, we observe substantial bias across all five estimators. Nonetheless, the MII estimator attains the lowest bias and maintains the lowest variance, reinforcing its position as the most effective method in this setting. This observation further validates our choice of assigning moderate and balanced weights to the outcome components. Moreover, AMII continues to achieve performance comparable to MII, even when paired with a GNN architecture that is not carefully designed.

In conclusion, we find that MII performs best when there is no interaction term and thus no associated discrepancy issue. In addition, AMII demonstrates competitive performance even with a simple GNN. Since real-world scenarios almost always involve complex interaction effects, and the clean structure of Equation (20) is idealized, we recommend using AMII in practice whenever it is feasible to train a counterfactual predictor.

### A.3 IMPACT OF TRAINING SAMPLE SELECTION

In this subsection, we investigate whether training the counterfactual predictor on the full population or solely on the boundary units yields better performance. This question naturally arises from the idea of sample splitting, although the intricate interdependencies within the data cannot be fully eliminated by this approach. We report the results in Table 12 and 13, and compare them to Table 3 and Table 6. We observe that training the counterfactual predictor solely on boundary nodes performs worse when $p = 0.1$, but becomes slightly better when $p = 0.3, 0.5$. The underlying intuition is that boundary nodes become more informative as $p$ increases, since the exposure—e.g., the proportion of treated neighbors—shifts toward the global treatment regime. Nevertheless, the difference between these two approaches remains minor.

### A.4 IMPACT OF NETWORK TOPOLOGY

Finally, we conduct an additional experiment on a different network topology[3] with a comparable scale but sparser connectivity—specifically, its average degree is approximately half that of the network used in our earlier experiments. Although the potential outcome model remains the primary focus of our simulation study, we include this experiment for completeness. The detailed results are presented in Tables 14 and 15, alongside comparisons with Tables 3 and 6.

We observe that the performance of the other three estimators—which lack a regression component—remains largely unaffected by changes in the network topology. Additionally, the GNN-based counterfactual prediction estimator demonstrates improved performance on this sparser graph, and this enhancement consistently translates to better performance for the AMII estimator as well.

Table 2: Statistical performance ($r_2 = 0, \gamma = 2$)

| $p$ Metric Estimators | 0.1 Bias | Std | MSE | 0.3 Bias | Std | MSE | 0.5 Bias | Std | MSE |
|---|---|---|---|---|---|---|---|---|---|
| **Hájek** | 0.925 | 0.405 | 1.020 | 0.871 | 0.289 | 0.842 | 0.828 | 0.197 | 0.725 |
| **CAE** | 0.942 | 0.455 | 1.094 | 0.940 | 0.237 | 0.940 | 0.923 | 0.159 | 0.878 |
| **MII** | 0.947 | 0.302 | 0.987 | 0.944 | 0.130 | 0.908 | 0.940 | 0.092 | 0.892 |
| **GNN** | 1.796 | 1.070 | 4.369 | 0.632 | 0.216 | 0.446 | 0.429 | 0.126 | 0.199 |
| **AMII** | 0.694 | 0.465 | 0.699 | 0.265 | 0.169 | 0.099 | 0.187 | 0.100 | 0.045 |

---

[3]The network data is available at https://networkrepository.com/socfb-USF51.php.

Table 3: Statistical performance ($r_2 = 0$ , $\gamma = 5$)

| $p$ |  | 0.1 |  |  | 0.3 |  |  | 0.5 |  |
| Metric |  |  |  |  |  |  |  |  |  |
| Estimators | Bias | Std | MSE | Bias | Std | MSE | Bias | Std | MSE |
|---|---|---|---|---|---|---|---|---|---|
| **Hájek** | 0.965 | 0.558 | 1.243 | 0.908 | 0.420 | 1.001 | 0.874 | 0.370 | 0.901 |
| **CAE** | 0.947 | 0.342 | 1.014 | 0.948 | 0.193 | 0.937 | 0.948 | 0.136 | 0.917 |
| **MII** | 0.955 | 0.242 | 0.971 | 0.961 | 0.130 | 0.940 | 0.961 | 0.092 | 0.933 |
| **GNN** | 1.497 | 0.997 | 3.235 | 0.650 | 0.173 | 0.452 | 0.430 | 0.114 | 0.198 |
| **AMII** | 0.597 | 0.412 | 0.526 | 0.249 | 0.160 | 0.088 | 0.165 | 0.097 | 0.037 |

Table 4: Statistical performance ($r_2 = 0$, $\gamma = 10$)

| $p$ |  | 0.1 |  |  | 0.3 |  |  | 0.5 |  |
| Metric |  |  |  |  |  |  |  |  |  |
| Estimators | Bias | Std | MSE | Bias | Std | MSE | Bias | Std | MSE |
|---|---|---|---|---|---|---|---|---|---|
| **Hájek** | 0.954 | 0.593 | 1.261 | 0.929 | 0.466 | 1.081 | 0.890 | 0.400 | 0.952 |
| **CAE** | 0.982 | 0.349 | 1.086 | 0.981 | 0.192 | 0.999 | 0.968 | 0.140 | 0.956 |
| **MII** | 0.975 | 0.241 | 1.010 | 0.974 | 0.129 | 0.965 | 0.971 | 0.098 | 0.952 |
| **GNN** | 1.608 | 1.009 | 3.603 | 0.678 | 0.169 | 0.489 | 0.448 | 0.110 | 0.213 |
| **AMII** | 0.642 | 0.401 | 0.572 | 0.242 | 0.147 | 0.080 | 0.153 | 0.101 | 0.034 |

Table 5: Statistical performance ($r_2 = 1$, $\gamma = 2$)

| $p$ |  | 0.1 |  |  | 0.3 |  |  | 0.5 |  |
| Metric |  |  |  |  |  |  |  |  |  |
| Estimators | Bias | Std | MSE | Bias | Std | MSE | Bias | Std | MSE |
|---|---|---|---|---|---|---|---|---|---|
| **Hájek** | 1.396 | 0.428 | 2.133 | 1.202 | 0.310 | 1.541 | 1.061 | 0.217 | 1.173 |
| **CAE** | 1.466 | 0.459 | 2.360 | 1.352 | 0.243 | 1.888 | 1.227 | 0.168 | 1.533 |
| **MII** | 1.422 | 0.329 | 2.131 | 1.294 | 0.149 | 1.697 | 1.200 | 0.103 | 1.451 |
| **GNN** | 2.247 | 1.170 | 6.418 | 1.101 | 0.228 | 1.263 | 0.770 | 0.157 | 0.618 |
| **AMII** | 1.150 | 0.492 | 1.563 | 0.713 | 0.192 | 0.545 | 0.486 | 0.124 | 0.251 |

Table 6: Statistical performance ($r_2 = 1$ , $\gamma = 5$)

| $p$ |  | 0.1 |  |  | 0.3 |  |  | 0.5 |  |
| Metric |  |  |  |  |  |  |  |  |  |
| Estimators | Bias | Std | MSE | Bias | Std | MSE | Bias | Std | MSE |
|---|---|---|---|---|---|---|---|---|---|
| **Hájek** | 1.480 | 0.575 | 2.520 | 1.274 | 0.438 | 1.814 | 1.129 | 0.377 | 1.416 |
| **CAE** | 1.555 | 0.345 | 2.538 | 1.428 | 0.195 | 2.076 | 1.293 | 0.138 | 1.692 |
| **MII** | 1.471 | 0.274 | 2.239 | 1.341 | 0.154 | 1.822 | 1.239 | 0.100 | 1.544 |
| **GNN** | 1.917 | 0.906 | 4.494 | 1.142 | 0.175 | 1.336 | 0.792 | 0.117 | 0.641 |
| **AMII** | 1.077 | 0.388 | 1.311 | 0.731 | 0.168 | 0.563 | 0.480 | 0.100 | 0.241 |

Table 7: Statistical performance ($r_2 = 1$, $\gamma = 10$)

| $p$ |  | 0.1 |  |  | 0.3 |  |  | 0.5 |  |
| Metric |  |  |  |  |  |  |  |  |  |
| Estimators | Bias | Std | MSE | Bias | Std | MSE | Bias | Std | MSE |
|---|---|---|---|---|---|---|---|---|---|
| **Hájek** | 1.493 | 0.605 | 2.597 | 1.319 | 0.476 | 1.965 | 1.164 | 0.408 | 1.522 |
| **CAE** | 1.628 | 0.353 | 2.775 | 1.486 | 0.193 | 2.246 | 1.334 | 0.141 | 1.798 |
| **MII** | 1.514 | 0.264 | 2.362 | 1.379 | 0.141 | 1.921 | 1.271 | 0.103 | 1.625 |
| **GNN** | 1.979 | 0.897 | 4.722 | 1.184 | 0.176 | 1.433 | 0.824 | 0.111 | 0.691 |
| **AMII** | 1.131 | 0.395 | 1.434 | 0.753 | 0.150 | 0.590 | 0.489 | 0.100 | 0.249 |

Table 8: Statistical performance **with one covariate** ($r_2 = 0, \gamma = 5$)

| $p$ | | 0.1 | | | 0.3 | | | 0.5 | |
| **Metric** | Bias | Std | MSE | Bias | Std | MSE | Bias | Std | MSE |
| **Estimators** | | | | | | | | | |
| **Hájek** | 0.958 | 0.557 | 1.228 | 0.905 | 0.416 | 0.992 | 0.870 | 0.366 | 0.890 |
| **CAE** | 0.935 | 0.342 | 0.992 | 0.936 | 0.193 | 0.913 | 0.933 | 0.136 | 0.889 |
| **MII** | 0.949 | 0.242 | 0.959 | 0.957 | 0.130 | 0.933 | 0.958 | 0.092 | 0.927 |
| **GNN** | 1.428 | 0.991 | 3.023 | 0.639 | 0.173 | 0.439 | 0.425 | 0.116 | 0.195 |
| **AMII** | 0.578 | 0.394 | 0.490 | 0.274 | 0.160 | 0.101 | 0.197 | 0.100 | 0.049 |

Table 9: Statistical performance **with one covariate** ($r_2 = 1, \gamma = 5$)

| $p$ | | 0.1 | | | 0.3 | | | 0.5 | |
| **Metric** | Bias | Std | MSE | Bias | Std | MSE | Bias | Std | MSE |
| **Estimators** | | | | | | | | | |
| **Hájek** | 1.472 | 0.572 | 2.493 | 1.271 | 0.431 | 1.800 | 1.124 | 0.371 | 1.401 |
| **CAE** | 1.544 | 0.345 | 2.502 | 1.415 | 0.195 | 2.040 | 1.279 | 0.138 | 1.654 |
| **MII** | 1.465 | 0.270 | 2.218 | 1.337 | 0.151 | 1.811 | 1.236 | 0.099 | 1.536 |
| **GNN** | 1.857 | 0.876 | 4.215 | 1.139 | 0.160 | 1.322 | 0.785 | 0.108 | 0.627 |
| **AMII** | 1.076 | 0.377 | 1.301 | 0.754 | 0.161 | 0.595 | 0.507 | 0.097 | 0.267 |

Table 10: Statistical performance **without covariates** ($r_2 = 0, \gamma = 5$)

| $p$ | | 0.1 | | | 0.3 | | | 0.5 | |
| **Metric** | Bias | Std | MSE | Bias | Std | MSE | Bias | Std | MSE |
| **Estimators** | | | | | | | | | |
| **Hájek** | 0.019 | 0.558 | 0.312 | 0.001 | 0.415 | 0.172 | 0.013 | 0.369 | 0.136 |
| **CAE** | 0.023 | 0.342 | 0.118 | 0.016 | 0.193 | 0.038 | 0.008 | 0.136 | 0.019 |
| **MII** | 0.011 | 0.242 | 0.059 | 0.003 | 0.130 | 0.017 | 0.002 | 0.092 | 0.009 |
| **GNN** | 1.708 | 0.555 | 3.224 | 0.561 | 0.148 | 0.337 | 0.389 | 0.089 | 0.159 |
| **AMII** | 0.011 | 0.264 | 0.070 | 0.058 | 0.177 | 0.035 | 0.034 | 0.118 | 0.015 |

Table 11: Statistical performance **without covariates** ($r_2 = 1, \gamma = 5$)

| $p$ | | 0.1 | | | 0.3 | | | 0.5 | |
| **Metric** | Bias | Std | MSE | Bias | Std | MSE | Bias | Std | MSE |
| **Estimators** | | | | | | | | | |
| **Hájek** | 0.533 | 0.572 | 0.611 | 0.366 | 0.427 | 0.316 | 0.268 | 0.371 | 0.209 |
| **CAE** | 0.586 | 0.345 | 0.463 | 0.463 | 0.194 | 0.252 | 0.338 | 0.138 | 0.133 |
| **MII** | 0.504 | 0.270 | 0.327 | 0.377 | 0.151 | 0.165 | 0.275 | 0.099 | 0.086 |
| **GNN** | 2.397 | 0.792 | 6.373 | 1.098 | 0.150 | 1.227 | 0.771 | 0.109 | 0.607 |
| **AMII** | 0.552 | 0.284 | 0.385 | 0.464 | 0.176 | 0.247 | 0.309 | 0.120 | 0.110 |

Table 12: Statistical performance with **regression on boundary** ($r_2 = 0, \gamma = 5$)

| $p$ | | 0.1 | | | 0.3 | | | 0.5 | |
| **Metric** | Bias | Std | MSE | Bias | Std | MSE | Bias | Std | MSE |
| **Estimators** | | | | | | | | | |
| **Hájek** | 0.965 | 0.558 | 1.243 | 0.908 | 0.420 | 1.001 | 0.874 | 0.370 | 0.901 |
| **CAE** | 0.947 | 0.342 | 1.014 | 0.948 | 0.193 | 0.937 | 0.948 | 0.136 | 0.917 |
| **MII** | 0.955 | 0.242 | 0.971 | 0.961 | 0.130 | 0.940 | 0.961 | 0.092 | 0.933 |
| **GNN** | 2.281 | 1.030 | 6.265 | 0.662 | 0.167 | 0.466 | 0.446 | 0.118 | 0.213 |
| **AMII** | 0.740 | 0.421 | 0.725 | 0.144 | 0.204 | 0.062 | 0.092 | 0.131 | 0.026 |

Table 13: Statistical performance with **regression on boundary** ($r_2 = 1, \gamma = 5$)

| $p$ Metric Estimators | 0.1 Bias | Std | MSE | 0.3 Bias | Std | MSE | 0.5 Bias | Std | MSE |
|---|---|---|---|---|---|---|---|---|---|
| **Hájek** | 1.480 | 0.575 | 2.520 | 1.274 | 0.438 | 1.814 | 1.129 | 0.377 | 1.416 |
| **CAE** | 1.555 | 0.345 | 2.538 | 1.428 | 0.195 | 2.076 | 1.293 | 0.138 | 1.692 |
| **MII** | 1.471 | 0.274 | 2.239 | 1.341 | 0.154 | 1.822 | 1.239 | 0.100 | 1.544 |
| **GNN** | 2.463 | 1.211 | 7.535 | 1.167 | 0.190 | 1.398 | 0.812 | 0.117 | 0.673 |
| **AMII** | 1.080 | 0.492 | 1.408 | 0.625 | 0.222 | 0.440 | 0.386 | 0.136 | 0.167 |

Table 14: Statistical performance on FB-USF ($r_2 = 0, \gamma = 5$)

| $p$ Metric Estimators | 0.1 Bias | Std | MSE | 0.3 Bias | Std | MSE | 0.5 Bias | Std | MSE |
|---|---|---|---|---|---|---|---|---|---|
| **Hájek** | 0.906 | 0.646 | 1.238 | 0.794 | 0.508 | 0.889 | 0.710 | 0.435 | 0.694 |
| **CAE** | 0.954 | 0.311 | 1.007 | 0.925 | 0.165 | 0.883 | 0.896 | 0.109 | 0.815 |
| **MII** | 0.944 | 0.241 | 0.950 | 0.943 | 0.131 | 0.906 | 0.943 | 0.101 | 0.899 |
| **GNN** | 0.821 | 0.381 | 0.819 | 0.559 | 0.187 | 0.347 | 0.346 | 0.094 | 0.128 |
| **AMII** | 0.302 | 0.256 | 0.157 | 0.158 | 0.138 | 0.044 | 0.144 | 0.101 | 0.031 |

Table 15: Statistical performance on FB-USF ($r_2 = 1, \gamma = 5$)

| $p$ Metric Estimators | 0.1 Bias | Std | MSE | 0.3 Bias | Std | MSE | 0.5 Bias | Std | MSE |
|---|---|---|---|---|---|---|---|---|---|
| **Hájek** | 1.432 | 0.651 | 2.475 | 1.188 | 0.511 | 1.672 | 0.989 | 0.442 | 1.173 |
| **CAE** | 1.549 | 0.314 | 2.498 | 1.400 | 0.171 | 1.988 | 1.253 | 0.117 | 1.583 |
| **MII** | 1.483 | 0.250 | 2.261 | 1.364 | 0.138 | 1.881 | 1.259 | 0.106 | 1.597 |
| **GNN** | 1.457 | 0.366 | 2.258 | 1.001 | 0.142 | 1.021 | 0.714 | 0.094 | 0.518 |
| **AMII** | 0.879 | 0.281 | 0.851 | 0.633 | 0.137 | 0.419 | 0.504 | 0.105 | 0.265 |

# B  PROOF

## B.1  PROOF OF THEOREM 3.1

First, we introduce the basic notations: $m_k = |\operatorname{Int}_k|$ denotes the size of the interior set of $k$-th cluster, $n_k = |C_k|$ denotes the size of $k$-th cluster. Moreover, we use $\bar{Y}_{k,Int}(\mathbf{1}), \bar{Y}_{k,Int}(\mathbf{0})$ denote the mean outcome of the interior nodes of the $k$-th cluster, under global treatment and control, respectively. Correspondingly, $\bar{Y}_k(\mathbf{1}), \bar{Y}_k(\mathbf{0})$ denote the mean outcome of the nodes of the $k$-th cluster, under global treatment and control, respectively. At last, we use $t_k$ as the treatment assignment indicator for the $k$-th cluster.

Since we employ cluster-level randomization, the treatment assignments of units within the same cluster are highly correlated. Under the neighborhood interference assumption, we begin by rearranging our estimator as follows:

$$
\begin{aligned}
\hat{\tau}_{MII} &= \frac{\sum_{i\in\operatorname{Int}} z_i Y_i}{\sum_{j\in\operatorname{Int}} z_j} - \frac{\sum_{i\in\operatorname{Int}}(1-z_i)Y_i}{\sum_{j\in\operatorname{Int}}(1-z_j)} \\
&= \frac{\sum_{i\in\operatorname{Int}} z_i Y_i(\mathbf{1})}{\sum_{j\in\operatorname{Int}} z_j} - \frac{\sum_{i\in\operatorname{Int}}(1-z_i)Y_i(\mathbf{0})}{\sum_{j\in\operatorname{Int}}(1-z_j)} \\
&= \frac{\sum_{k\in[K]}\sum_{i\in\operatorname{Int}_k} z_i Y_i(\mathbf{1})}{\sum_{j\in\operatorname{Int}} z_j} - \frac{\sum_{k\in[K]}\sum_{i\in\operatorname{Int}_k}(1-z_i)Y_i(\mathbf{0})}{\sum_{j\in\operatorname{Int}}(1-z_j)} \\
&= \frac{\sum_{k\in[K]} t_k m_k \bar{Y}_{k,Int}(\mathbf{1})}{\sum_{k\in[K]} t_k m_k} - \frac{\sum_{k\in[K]}(1-t_k)m_k \bar{Y}_{k,Int}(\mathbf{0})}{\sum_{k\in[K]}(1-t_k)m_k}.
\end{aligned}
\tag{21}
$$

The second equality holds due to NIA (Assumption 2.1) and cluster-level randomization. The third equality is transforming the summation over all interior units to that first over interior of $k$-th cluster, then over clusters. The fourth equality is definition.

By applying Slutsky's lemma, we eliminate the randomness in treatment assignments, yielding:

$$
\hat{\tau}_{MII} = \frac{\sum_{k\in[K]} m_k \bar{Y}_k(\mathbf{1})}{\sum_{k\in[K]} m_k} - \frac{\sum_{k\in[K]} m_k \bar{Y}_k(\mathbf{0})}{\sum_{k\in[K]} m_k} + o_p(1).
\tag{22}
$$

Assumption 3.1 is adapted from the Assumptions 4.1.1 and 4.1.2 from the original paper of CAE (Liu et al., 2024), while ours are weaker partly because we only pursue consistency. Given Assumption 3.1, we can substitute the component of interior nodes into that of the whole cluster, yielding:

$$
\begin{aligned}
\hat{\tau}_{MII} &= \frac{\sum_{k\in[K]} n_k \bar{Y}_{k,Int}(\mathbf{1})}{\sum_{k\in[K]} n_k} - \frac{\sum_{k\in[K]} n_k \bar{Y}_{k,Int}(\mathbf{0})}{\sum_{k\in[K]} n_k} + o_p(1) \\
&= \frac{\sum_{k\in[K]} n_k \bar{Y}_k(\mathbf{1})}{\sum_{k\in[K]} n_k} - \frac{\sum_{k\in[K]} n_k \bar{Y}_k(\mathbf{0})}{\sum_{k\in[K]} n_k} + o_p(1) \\
&= \tau + o_p(1).
\end{aligned}
\tag{23}
$$

## B.2  PROOF OF THEOREM 4.1

First, we define $h(e) = \mathbb{E}_v[h(e,v)]$. Recall that the potential outcome model is:

$$
Y_i(\mathbf{z}) = (\beta + \alpha u_i)z_i + h\left(\sum_{j\in\mathcal{N}(i,1)} z_j/\deg_i, v_i\right).
\tag{24}
$$

Thus, the true GATE is given by:

$$
\tau = \beta + \alpha\mu + h(1) - h(0).
\tag{25}
$$

Next, with this model, the expectation of MII estimator is given by:

$$
\hat{\tau}_{MII} = \beta + \alpha\mu_{Int} + h(1) - h(0).
\tag{26}
$$

These together give:

$$\text{Bias}(\hat{\tau}_{MII}) = \alpha(\mu - \mu_{Int}). \tag{27}$$

Next, we examine the AMII estimator. We define:

$$\hat{h}(p) = \mathbb{E}_{\mathbf{z}\sim\text{Ber}(p),v}[\text{MEAN}(\{g(z_j, v_j) \mid j \in \mathcal{N}(i,1)\})]. \tag{28}$$

This represents the expectation of the estimate of the interference part given the treatment proportion of randomization equal to $p$. Notice that the randomness lies in both treatment assignment and the covariate $v$. Nonetheless, the former diminish when $p = 0$ or $p = 1$, since wherein $\mathbf{z} = \mathbf{0}, \mathbf{1}$ holds, respectively.

An important trait of the functional form of regression function lies in:

$$\hat{h}(1) = \mathbb{E}_v \left[ \frac{1}{|\text{Int}|} \sum_{i\in\text{Int}} \text{MEAN}(\{g(1, v_j) \mid j \in \mathcal{N}(i,1)\}) \middle| \mathcal{G} \right]. \tag{29}$$

This holds because one can swap the MEAN and expectation $\mathbb{E}_v$, given the degree $\deg_i$ fixed. On the other hand, one can easily verify:

$$\hat{h}(1) = \mathbb{E}_v \left[ \frac{1}{n} \sum_{i\in[n]} \text{MEAN}(\{g(1, v_j) \mid j \in \mathcal{N}(i,1)\}) \middle| \mathcal{G} \right]. \tag{30}$$

Thus, we have:

$$\begin{aligned}
\mathbb{E}[\hat{\tau}_{AMII}] &= \beta - \mathbb{E}[\hat{\beta}_n] + (\alpha - \mathbb{E}[\hat{\alpha}_n])\mu_{Int} + h(1) - \hat{h}(1) + \mathbb{E}[\hat{\beta}_n] + \mathbb{E}[\hat{\alpha}_n]\mu + \hat{h}(1) \\
&\quad - \left(h(0) - \hat{h}(0) + \hat{h}(0)\right) \\
&= \beta + (\alpha - \mathbb{E}[\hat{\alpha}_n])\mu_{Int} + \mathbb{E}[\hat{\alpha}_n]\mu + h(1) - h(0).
\end{aligned} \tag{31}$$

This gives the bias of AMII estimator:

$$\text{Bias}(\hat{\tau}_{AMII}) = (\mathbb{E}[\hat{\alpha}] - \alpha)(\mu - \mu_{Int}). \tag{32}$$

