# OpenReview forum: "Journey to the Centre of Cluster: Harnessing Interior Nodes for A/B Testing under Network Interference"
_ICLR.cc/2026/Conference — ICLR 2026 Poster_

### Official Review · Reviewer_RLAt · 2025-10-27

**Soundness:** 2
**Presentation:** 1
**Contribution:** 2
**Rating:** 2
**Confidence:** 4

**Summary:**

This paper addresses the challenge of conducting A/B tests on networks where interference between connected units violates standard causal assumptions. The authors propose the Mean-in-Interior (MII) estimator, which leverages interior nodes lying within the same cluster and *eliminates the inefficiency* introduced by the cluster-relevant weights used in the previous CAE estimator. Recognizing that interior nodes may not represent the overall population, the paper further introduces the Augmented MII (AMII) estimator, which employs a GNN-based counterfactual predictor to correct for selection bias. Theoretical analysis implies that AMII has an intuitively smaller bias than MII estimator in specific additive model, with another interpretation based on semi-supervised adjustment provided in terms of this method. Extensive simulations on semi-synthetic social network data demonstrate that AMII generally achieves lower MSE than comparative methods, establishing it as an efficient and practical approach for causal estimation under network interference.

**Strengths:**

The paper introduces the Mean-in-Interior (MII) and Augmented MII (AMII) estimators, fulfilling the gaps in GATE estimation based on interior nodes, and draws the attention on possible incomplete representativeness of the interior nodes to the entire cluster.

**Weaknesses:**

The paper features several notable weaknesses. While the authors attempt to convey dense ideas through multiple key intuitions and assertions, few are elaborated with sufficient rigor to enable non-expert readers to fully grasp them. The theoretical justification for the advantages of the MII and AMII estimators remains limited. Moreover, the rationale for viewing the MII estimator as an improvement over the CAE estimator is insufficient and not theoretically supported. Additionally, Section 3, which introduces the MII estimator, and Section 4, which discusses covariate adjustment, appear insufficiently connected in their conceptual development. Please see Questions for details.

**Questions:**

1. **On the novelty of the MII estimator**
   Overall, I think the novelty of the proposed MII estimator could be better explained. From what I can tell, it’s basically a difference-in-means estimator applied to interior nodes, which is a pretty classical idea in causal inference. Also, the idea of leveraging interior nodes has been seen in earlier work. It would help if the authors could clarify the genuine improvement on MII estimator here in rigorous theories.

2. **On theoretical justification for the advantage of MII over CAE**
   In lines 250–262, the paper argues that MII improves efficiency over the CAE estimator by using better cluster-relevant weights. But CAE’s weighting scheme is rooted in **cluster randomization**, which doesn’t seem to be part of the main assumptions (2.1 and 3.1). So, it feels like the two estimators might be targeting different experimental designs, making the motivation a bit insufficient. Also, there doesn’t seem to be a clear theoretical guarantee backing this claim—some clarification would be great.

3. **On clarity of statements**
   A few parts could use clearer explanations or more precise wording:
   - *Line 48:* The paper says that leveraging the known graph structure is *much more appealing*, but it doesn’t really argue why.
   - *Line 251:* I didn’t quite follow why the model *helps build intuition* that variance analysis is more complex than bias analysis—or why it’s necessary to emphasize that intuition here.
   - *Lines 315–316:* The statement *they may struggle … a comparative advantage* is confusing. It’s not clear why it “struggles,” or why an $f$ -model that doesn’t estimate the interference function $h$ should be considered an advantage. Since any model $f$ can adapt, this doesn’t seem to tie closely to the covariate adjustment goal.
   - *Line 275-276:* *We argue that ... can be violated.* It does not provide any explanation not state exactly which assumptions are violated.
   - *Line 281-282:* *Most estimators without a regression component tend to rely heavily on interior nodes*, there have been a number of estimators on exposure effects that do not rely on interior nodes as well, thus an arbitrary assertion.
   - The inconsistent names of $h$ as both ''transformation function'' (line 255) and ''interference function'' (line 315).

4. **On covariate adjustment and coherence**
   Covariate adjustment can be applied to pretty much any estimator, not just MII. So it would be fairer to compare **AMII** with the augmented versions of all other estimators too. Also, Sections 3 and 4 don’t quite feel conceptually aligned—cluster-related weights and covariate adjustment seem like two separate issues, and it might help to connect them more clearly.

5. **On Assumption 4.1 and model training**
   Assumption 4.1 says that $f$ is trained in a regression form, but in line 290 the paper says $f$ is trained using a GNN. How does that assumption hold in this case? A short explanation would make this part clearer.

6. **On experiments and ablation studies**
   The experimental section feels a bit thin. It would really help to include a few ablation studies, for example:
   - What happens if covariates actually affect the outcome?
   - What if the interior nodes aren’t very representative of the full cluster?
   Seeing these variations would strengthen the experimental support for MII.

**Details Of Ethics Concerns:**

No concerns on ethnics.

---

> ### Author Response · Authors · 2025-11-13
>
> We sincerely thank the reviewer for their valuable effort in reviewing our paper. We hope to clarify several issues and address their concerns below.
>
> First, we wish to clarify that our statements are grounded in **empirical observations and numerical instances, not pure intuition**. Many of these points highlight **practical limitations of existing methods** that we identified, which we then **illustrate using a concrete running example** in the paper.
>
> In addition to the detailed responses below, if the reviewer has concerns about any other specific statements, we warmly invite them to specify these so we may provide further clarification.
>
>
>
> ## Comparison of MII estimator and CAE (Novelty of MII estimators)
>
> In Section 4.1, we argue that the clustering algorithm used in practice cannot depend on the **outcomes** of units; thus, the assumptions imposed on cluster-level outcomes are fragile. Consequently, we claim that the **cluster-aware weights used in CAE are unnecessary**, and we instead adopt equal weights in the MII estimator.
>
> The theoretical justification for the lower variance of MII is based on the potential outcome model in Equation (11), as briefly stated in Lines 260–261 (due to space constraints). Specifically, when each $Y_i$ is an unbiased estimate of a common parameter $\theta = E[Y_i]$, the estimator $\hat\theta = \sum_{i=1}^m w_i Y_i$ achieves minimum variance among all unbiased estimators when $w_i = 1/m, \forall i \in [m]$, as justified by the Cauchy–Schwarz inequality. Introducing unnecessary weights—such as the bi-level weights used in CAE—would increase variance in this setting.
>
> Furthermore, we restrict the available samples to **interior nodes**, since they always yield unbiased estimates of the mean outcomes under global treatment/control, given Equation (13). Including outcomes of **boundary nodes** could reduce variance but at the cost of introducing bias. However, as illustrated in the introduction, boundary nodes contribute little to the estimator, including CAE. Hence, we provide the insight that MII achieves lower variance when the cluster structure is weak (we will clarify what “weak” means in our response to Question 2). Importantly, both our MII estimator and CAE are consistent under similar representativeness assumptions. Therefore, while their biases are comparable, the MII estimator attains lower variance.
>
> As illustrated in our simulation study (in detail, Tables 2 to 15), MII always achieves substantially **lower variance** (due to equal weights) while achieving bias similar to CAE. We also prove that, with similar assumptions, both CAE and MII estimators are consistent, which explains their similar behavior in terms of bias.
>
> Finally, we clarify why boundary nodes contribute negligibly to CAE. For instance, when $p=10%$ (which corresponds to the intermediate treatment phase in practice), there are 102 fully treated nodes, among which only 7 are boundary nodes (this result is an expectation based on Monte Carlo simulations, rounded up). The HT, CAE, and Hájek estimators all retain only these fully treated nodes.
>
>
>
>
>
> ### Novelty
>
> While the difference-in-means estimator is indeed classic, our MII estimator is nontrivial. Its novelty stems from **applying this estimator to a subtle, carefully chosen network-aware subgroup—the interior nodes.** This selection is directly motivated by the context of network interference and is, to our knowledge, a novel proposal in this literature.
>
> We also wish to highlight a crucial challenge within this domain: proving a **general, analytical improvement** of one estimator over another is often **intractable** in the existing literature on GATE estimation under network interference. For example, Theorem 4.2 in *Liu et al. (2024)* shows that CAE is consistent, but its asymptotic variance expression is highly complex, involving numerous high-level, network-relevant nuisance parameters that are themselves often unidentifiable. We must clarify that variance analysis under network interference is notoriously complex. Even when (and if) coarse asymptotic variance expressions are derived, a **direct analytical comparison between two such expressions is generally intractable**, as far as we are aware.
>
> Hence, our approach is to leverage the classic and general potential outcome model in Equation (13) to **provide a principled justification** for using equal weights (MII) over the complex, cluster-aware bi-level weights (CAE).

---

> ### Author Response · Authors · 2025-11-13
>
> ## Question 2
>
> First, we clarify that both MII and CAE are designed for **cluster-level randomization**, which is a standard setting in industrial applications and the relevant methodological literature.
>
> Next, we must clarify a crucial point: CAE's bi-level weighting does **not** stem from the cluster-level randomization itself. Instead, it is a direct consequence of a **strong structural assumption**—namely, that all clusters share the **same** within-cluster mean potential outcome under global treatment or control (see Assumption 4.3.1 in *Liu et al. (2024)* for reference).
>
> Our discussion at the beginning of Section 4 argues that, due to computational constraints, the clustering algorithms used in practice are typically heuristic and **downstream-agnostic**. Consequently, they produce "weak" cluster structures, not the "strong" structures implicitly required by CAE's assumption.
>
> This reflects our observations from industrial practice. For instance, two of our main collaborators, working with social networks at the scale of hundreds of millions of users, utilize the Louvain and Leiden algorithms. This choice is driven by operational necessity: clustering must be updated at least weekly, mandating the use of low-complexity algorithms. Our core point is that the cluster structures generated by these common, fast, and heuristic algorithms **cannot be expected to satisfy the strong outcome-homogeneity assumption** that CAE relies upon for its weighting scheme.
>
>
>
>
>
>
>
>
>
> ## Question 3 (Clarification of statements)
>
> ### Line 48
>
> This problem is a bit nuanced, so we omit detailed discussion, but the high-level idea is brief: if one has access to abundant additional information, i.e., the whole network topology, they undoubtedly have more potential to solve the problem better.
>
> To be more specific, the reason is that a graph-agnostic method would face a very challenging extrapolation task. For example, given a specific randomization scheme, one can view the mean outcomes as a 1-dimensional function of treatment proportion, e.g., $M(p)$. The GATE can be written as $M(1)-M(0)$. However, in practice, most experiments are allocated with small traffic, which can be interpreted as $p=0.01$, e.g., refer to *SQR: Balancing Speed, Quality and Risk in Online Experiments*. Even with more traffic, we have access to $M$ at points $p=0.01, 0.02, 0.05$, but our target scenario is $p=1$, which is a highly risky extrapolation.
>
> Nonetheless, even with $p=0.05$, if cluster-level randomization is applied and we admit NIA (Assumption 3.1), there can be a few units that lie in an environment similar to that of desired global treatments, but this can be captured only through the concrete network topology.
>
>
>
>
>
> ### Line 251
>
> We again thank the reviewer for their careful reading. We acknowledge that it was our oversight not to create a new line for the second sentence at Line 251. The purpose of Equation (13) is to build intuition on **why the MII estimator achieves lower variance compared to other weighted estimators**. To improve readability, we will separate the two sentences at Lines 250–251 into two paragraphs in the revision.
>
> Additionally, we would like to clarify the rationale behind the complexity of the variance analysis. Under the potential outcome model in Equation (13), the variance of common estimators involves fourth-order terms of treatment vectors, e.g., $\operatorname{Cov}[z_i z_j, z_k z_l]$, as well as statistics of the network topology. This complexity is further compounded under cluster-level randomization, due to the interior/boundary attributes of nodes. For instance, $i,j,k,l$ may all lie in the same cluster or in different clusters, and the connections among them (represented by the adjacency matrix) must also be considered. For reference, please see *Theorem 1 in Chen & Li (2024)*, *Theorem 4.2 in Liu et al. (2024)*, and page 7 in the appendix of *Yu et al. (2022a)*. Although these variances (or asymptotic variances) can be written explicitly, they are often intractable to analyze or estimate in practice.
>
>
>
> ### Line 315-316
>
> Given NIA and cluster-level randomization, interior nodes always fall entirely within global treatment or global control conditions, eliminating the need to estimate any nuisance components—such as the interference function $h$ in Equation (13). However, the **micro-level regression function**, despite its potential advantages over graph-agnostic methods, would **struggle with extrapolation**, primarily due to the **low treatment proportion**. While boundary nodes constitute the vast majority of all nodes, their network exposure (e.g., the proportion of treated 1-hop neighbors in Equation (13)) tends to concentrate around $p$, which remains far below 1—the desired level under global treatment.
>
> For illustration, we employ a lightweight GNN model and refer the reviewer to the performance of the GNN method when $p=0.1$ in our simulation study (especially compared to that with $p=0.3,0.5$).

---

> ### Author Response · Authors · 2025-11-13
>
> ### Line 275-276
>
> The structural assumptions include Assumptions 4.3.1 and 4.3.2 in *Liu et al. (2024)* (CAE), which state that the mean outcomes within each cluster are identical (4.3.1) and that cluster-level representativeness holds (4.3.2, analogous to Assumption 3.1.2 in our paper, though we do not require a convergence rate for this gap). The common feature of these assumptions is that they impose structure on the **potential outcomes** within each cluster—a premise that is fragile in practice, given that clustering algorithms are typically **heuristic and downstream-agnostic**.
>
> In real-world applications, social networks with hundreds of millions of nodes are partitioned into millions of clusters, and it is unrealistic to expect that mean outcomes within each cluster are even approximately close; often, they may not exhibit concentration at all. Our key insight is that clustering in practice provides only a **weak structure**, insufficient to support strong assumptions on downstream outcomes—especially considering that a single experiment often involves multiple heterogeneous and simultaneously measured outcomes.
>
> In summary, we will add a clarifying sentence to explicitly explain the meaning of “structural assumptions” in this context.
>
> ### Line 281-282
>
> We clarify that our discussion consistently focuses on the **GATE estimator**, rather than the **exposure effect** considered in some related works.
>
> Moreover, the GATE estimators discussed in our paper—such as HT, Hájek, and CAE—heavily rely on **interior nodes**, particularly when the treatment proportion is low.
>
> Again, for instance, when $p = 10%$, there are 102 fully treated nodes, among which only 7 are boundary nodes. The HT, CAE, and Hájek estimators all retain only these fully treated nodes.
>
> ### Line 255 and 315
>
> We agree that this causes confusion, and we will unify the terminology as “interference function” in the next revision.
>
>
>
> ## Question 4
>
> We first propose the MII estimator in comparison to the CAE and HT estimators, and subsequently design the AMII estimator as an augmentation of MII (and for comparison with MII). Therefore, we do not believe this comparison is unfair. Moreover, while the three baseline methods can utilize boundary nodes, the MII estimator relies solely on interior nodes, and the AMII estimator again has access to boundary nodes.
>
> It is indeed feasible to add standard regression adjustments to the CAE or HT estimators.
> However, constructing a PPI-style augmentation—such as our AMII estimator—for CAE or HT is difficult to justify conceptually. This is because we select **interior nodes** as the “labeled” samples, which form a **fixed subgroup**determined by the network topology and clustering. In contrast, there is no analogous fixed subgroup for HT or CAE: their fully treated groups depend on the specific experimental design and the random realization of treatment assignments. Since the interior subgroup in MII is fixed, we can leverage predictions from boundary nodes to mitigate the systematic selection bias between boundary and interior units.
>
> Mathematically, the PPI-style estimator takes the form:
> $$\hat\tau_Y = \frac{1}{|\mathcal{D}|}\sum_{i\in\mathcal{D}} f(X_i) + \frac{1}{|\mathcal{D}_{\text{labeled}}|}\sum_{j\in \mathcal{D}_{\text{labeled}} } (Y_j - f(X_j))$$
> (There is unknown bug that this formula can not be compiled)
> However, for example, in the HT estimator, such a minus/plus adjustment changes nothing because all units can be incorporated in the HT estimator, according to its expression.

---

> ### Author Response · Authors · 2025-11-13
>
> ### Covariate adjustment issue
>
> This is a good point. In fact, our AMII estimator is not a simple regression adjustment but embodies a PPI point estimator. A classic one in causal inference is AIPW estimator. We compare our AMII with it for illustration.
>
> To begin, we claim that we only borrow the idea of the PPI estimator from a semi-supervised perspective, while our idea is different from not only PPI but also AIPW, especially due to the complexity of network interference compared to classic (i.i.d. or MACR) scenarios. In summary, our idea is **debiasing using prediction**, in comparison to the traditional **debiasing of the prediction**.
>
> First, the PPI estimator and the doubly robust estimator are important but different ones in semi-parametric inference. Our form embodies a **standard** point estimate of PPI, as presented in *Angelopoulos et al. (2023a)*, while, as discussed in *Angelopoulos et al. (2023b)*, in the task of mean estimation, the AIPW can be reorganized as a PPI++ estimator with **a particular choice of power tuner**. Moreover,  PPI estimator involves no nuisance estimator (AIPW involves two nuisance estimators) and behaves much differently from AIPW estimator in score space (as discussed in *Predictions as Surrogates: Revisiting Surrogate Outcomes in the Age of AI*).
>
> Second, concerning targets, the PPI-style estimator focuses more on the assumption-lean case and the prediction function $f$ is truly treated as an unknown black box, pursuing variance reduction as long as $f$ is correlated to $Y$. On the other hand, the AIPW estimator pursues consistency and semi-parametric efficiency when (at least one of) the two nuisances is **well-specified**.
>
> Finally, in our scenario, the data generation mechanism is much more complex beyond the classic i.i.d. data. For example, though we view the interior nodes as **labeled data**, the probability of a node becoming an interior node is usually too complicated to be calculated. We borrow the idea from PPI, while the story is actually different: even the sample mean of outcomes of interior nodes (MII estimator) can incur selection bias (mean of $Y$ in labeled data is unbiased in classic i.i.d. setting), and we **use prediction to mitigate the bias stemming from systematic covariate discrepancy between interior and boundary units**. The rationale is that the regression function can model covariates flexibly while incurring the difficulty of extrapolation of the treatment proportion.
>
>
>
>
> ### Section 3/4 alignment
>
> In our layout, Section 4.1 emphasizes two key points:
>
> - the clusters used in practice provide only a **weak structure**, and
> - there exists a potential discrepancy between the covariate distributions of interior and boundary nodes.
>
> Since we have shown that all network-aware estimators (including our MII estimator proposed in Section 3) primarily rely on interior nodes, a **selection bias** is therefore inevitable—as also validated empirically in Tables 2–9. To **mitigate**this systematic bias, we introduce a regression component.
>
> We will add a connecting sentence at the beginning of Section 4 in the next revision to improve the flow and presentation.
>
>
>
>
>
> ## Question 5
>
> In Line 290, we state that “for example, it may be instantiated as a GNN,” and indeed, we parameterize it as a GNN in our simulation study.
>
> Nonetheless, in **Assumption 4.1**, our goal is to illustrate that the regression function can effectively address covariate-related components. Based on the potential outcome model in Equation (13), we employ a **partial linear model**—comprising a linear interaction term and a graph convolution term—for illustration. In essence, **Assumption 4.1 and Theorem 4.1 are designed to demonstrate how AMII operates and performs** when systematic bias exists (i.e., when the representativeness condition in Assumption 3.1 is violated).
>
> In the phrase “one-layer graph convolution,” we mean that $g$ is a trainable function, such as a multilayer perceptron (MLP). We will add further explanation in the revision for clarity.
>
>
>
> ## Question 6
>
> In fact, we refer the reviewer to our Appendix A for our detailed results. Due to limited space, we mention that:
>
> > The complete experiments—including alternative resolution levels, different covariate specifications, different network topology, and a series of ablation studies—are reported in Appendix A. Here, we summarize the main conclusions from our results.
>
> in Line 429-430 and 452-453.
>
>
>
> Specifically, the two points raised by the reviewer actually refer to the same issue, as we model the **violation of the representativeness assumption (Assumption 3.1.2)** precisely through an **interaction term** between the treatment and a covariate that systematically differs between boundary and interior subgroups.
>
> In Figure 2 of the main paper, we introduce two such interaction terms (see Equation (17)). Additional covariate specifications are further explored in Section A.2 and Tables 8–11.

---

> > ### Comment · Reviewer_RLAt · 2025-11-27
> >
> > Thank the authors for their efforts. Given the improvement on the clarity of the paper’s contributions and key statements in the revised version, I will raise the score accordingly. Besides, I am still curious on the following questions.
> >
> > * In response to Question 3, it was mention that `The purpose of Equation (13) is to build intuition on why the MII estimator achieves lower variance compared to other weighted estimators. ` While the intuition is helpful, this key advantage would be significantly strengthened if a rigorous theorem could be established—even for simplified settings.
> > * In response to Question 4, the authors stated that the key idea of AMII estimator is to ` mitigate the bias between interior and boundary units.` Could you elaborate on the impact of the debiasing term on the estimator’s efficiency? For example, is there a notable bias–variance trade-off introduced by this adjustment?

---

> > > ### Author Response · Authors · 2025-11-28
> > >
> > > We sincerely thank the reviewer for recognizing our efforts!
> > >
> > > For question 3, we will add a new proposition in the ten-page version to explicitly elaborate on the efficiency gain of equal weights used in the MII estimator, based on the potential outcome model in Equation (14).
> > >
> > > For question 4, we first elaborate on the complexity of the variance analysis of the AMII estimator:
> > >
> > > - In a classic PPI setting (i.e., labeled and unlabeled data with missing at completely random), the prediction $\hat Y_i$ depends on $X_i $ solely, while in our setting, e.g., the prediction function is a GNN, the prediction $\hat Y_i$ can rely on the covariate (and treatment) of multiple units that involves network topology
> > > - In a classic PPI setting, the covariates $X_i$ are i.i.d., while in our setting, even the covariates are i.i.d., the other input, treatments, admits complex interdependency since we consider cluster-level randomization.
> > >
> > > Both these make the variance analysis intractable, so we can only characterize the bias with certain potential outcome models and an analytical form of the prediction function. Another example is Theorem 1 in Chen et al. (2024), where they exactly analyze the variance of the linear regression estimator and get an intricate expression.
> > >
> > > Next, we mention that in PPI++ Angelopoulos et al. (2023b), a power tuner is introduced to weight the adjustment term to minimize the variance (since the PPI point estimators are always unbiased in a classic setting, they can focus on variance reduction). Such a weight can also be put on the adjustment term in Equation (12) to invoke a bias-variance tradeoff, while we finally do not do so because it's very hard to construct a scientific rule to select that weight, and the reason is that the variance is very complex and we are unable to evaluate it through **single** experiment, even in simple settings. This is decided by the global trait of our estimand, GATE, since we have **no replicates** to evaluate the covariance of the MII estimator and the adjustment term. The variance of two parts and the covariance are usually needed for selecting such a weight, while they can only be evaluated through many repetitions of Monte Carlo simulation, which is infeasible in practice.
> > >
> > > Last, in our simulation studies, we deliberately increase the noise intensity (we set $\sigma=2$, which is equal to the magnitude of the direct effect) and find that our adjustment empirically loses some efficiency while achieving significant bias reduction. Hence, when the bias is the important one, the advantage of the AMII estimator is prominent, while when bias (specifically, the bias stemming from systematic discrepancy between interior and boundary units) is not significant, and the representativeness assumption holds, then the MII estimator is good enough.

---

### Official Review · Reviewer_vtCm · 2025-10-27

**Soundness:** 2
**Presentation:** 2
**Contribution:** 3
**Rating:** 4
**Confidence:** 3

**Summary:**

This paper proposes two methods for estimating the global average treatment effect (GATE) when the effect of a unit may depend on the treatment assignments of nearby units, with proximity defined by a graph structure over the units.

The first proposed method, Mean-In-Interior (MII) estimator, averages the outcomes of the interior nodes receiving the same treatment as that of each unit. The authors show that the MII estimator is consistent (Theorem 3.1) under the Neighborhood Interference Assumption (NIA; Assumption 2.1) and the assumption about the interior nodes (Assumption 3.1).
The authors also claim that the MII estimator reduces the variance compared to the recently proposed CAE estimator (Liu et al., 2024).

The second method, called Augmented MII (AMII), corrects the bias of the MII estimator due to the focus on the interior points, using a counterfactual predictor $f$, which predicts the expected outcome given specified treatment assignments of all units (as well as covariates and the graph). Theorem 4.1 presents the bias of this estimator under Assumption 4.1 about the model.

Finally, the paper presents experiments using a synthetic dataset with several specifications of the parameters of the data generating process. The results shown in Figure 2 demonstrate that the MII estimator has lower variance compared to the previous methods, and the AMII reduces the bias leading to reduced MSEs.

**Strengths:**

- The task of estimation of GATE in the presence of interference is practically relevant and challenging.

- The proposed methods are developed based on interesting ideas.

- Theorems 3.1 and 4.1 provide soundness and strength of the proposed methods.

I set the Contribution score 3: good because the direction of this work looks great.

**Weaknesses:**

- The proof of Theorem 4.1 (Appendix B.2) uses Eq. (24), which is only introduced "to illustrate the idea behind the AMII estimator". (The statement of Theorem 4.1 does not mention this.)

- I would need (at least) some more details about the Eq. (21) in the proof of Theorem 3.1 to verify the proof.

- It is unclear to me what Eq. (13) is illustrating.

- The AMII estimator looks like doubly robust/debiased estimators proposed in the standard treatment effect estimation literature, but there is no reference or discussion on this.

- The statement of Assumption 4.1 is not clear to me.

- The proposed methods are shown to work well on one dataset, which is a little underwhelming.

- The authors claim that the distributions in Figure 3 "exhibit clear discrepancies". However, it seems like a subjective observation and the criterion of this claim is not clear.

**Questions:**

Major concerns:
- Does the statement of each theorem include all of its required assumptions? For example, "treatment is assigned at the cluster level, meaning that all units within the same cluster receive the same treatment" (l.168): It is not clear if this is an assumption in the theory. Does Theorem 4.1 only needs Assumption 4.1? Please also see the Weaknesses section.

- Is Assumption 4.1 a condition about the true regression function, or the function used in the estimator? Is the function $g$ known to the estimator?

- Could you discuss the contributions in relation to doubly robust/debiased estimators?

- l. 227, "the MII estimator exhibits substantially lower variance": Why does the MII have lower variance compared to the CAE?

- I suppose the proposed estimators depends on the clusters through $\text{Int}_k$. Do the estimators fail if the clusters are not correct?


Minor issues:
- l.466, "In the absence of the interaction term and 2-hop interference, we find that MII is nearly unbiased": How did the author draw this observation?

- Is the difference between the distribution in Figure 3 statistically significant?

- Punctuation is missing after the equations.

- l.475, "increases exponentially": Isn't it polynomial in $p$?

- l.396, what is the difference between $\text{deg}_i$ and $c_i$?

---

> ### Author Response · Authors · 2025-11-13
>
> We sincerely thank the reviewer for their time and effort in carefully reviewing our paper. We clarify several points and address the raised concerns below.
>
>
>
> ## Use of Eq. (24)
>
> Indeed, we explicitly state that Equation (13) is required in **Theorem 4.1**:
>
> > “Given Assumption 4.1, the potential outcome model in Equation (13), ...”
>
> Equation (24) is identical to Equation (13); we restate it there solely for the reader’s convenience. In essence, the **parametric potential outcome model** is always necessary for conducting an exact bias analysis in the literature on network interference.
>
>
>
> ## Details of Eq. (21)
>
> First, we define the notation $\bar Y_{k,\text{Int}}(\cdot)$ at the beginning of the proof. Equation (21) is essentially a **reorganization of the summation**: in its original form, we sum over all interior units ($\sum_{i\in Int}$), while on the right-hand side, this summation is decomposed into two steps, $\sum_{k\in[K]}\sum_{j\in Int_k}$. Since we consider **cluster-level randomization**, all units within the $k$-th cluster receive the same treatment $t_k$. Consequently,
> $$
> t_k\sum_{j\in Int_k} Y_j = t_k m_k\bar Y_{k,Int} = t_k m_k\bar Y_{k,Int}(\mathbf{1})
> $$
> where the first equality follows from the definition, and the second equality follows from the **NIA** assumption—i.e., outcomes depend only on a unit’s own treatment and its 1-hop neighbors. Therefore, the outcomes of **treated interior nodes** are identical to those under global treatment.
>
> (Additionally, we have corrected a minor typo in the denominator summation in Equation (22) and (23): the summation index should be $\sum_{k \in [K]}$ rather than $\sum_{j \in Int}$.)
>
> Moreover, we will also add more detailed explanations in the next revision to make this step clearer.
>
>
>
>
>
>
>
> ## Meaning of Eq. (13)
>
> Literally, Equation (13) extends the classic form in Equation (11) by adding an **interaction term** between a covariate and the treatment, denoted as $u_i z_i$. For example, $u_i$ can represent the normalized degree, $\deg_i / \overline{\deg}$.
>
> This potential outcome model is designed to illustrate two key points:
>
> - The **limitation of statistical estimators without a regression component**—specifically, that systematic selection bias may arise when there is a substantial covariate discrepancy between interior and boundary nodes (as shown in Figure 3).
> - The **benefit of incorporating a regression component**, which can help mitigate this bias.
>
>
>
>
>
>
>
> ## Comparison with Doubly-robust (AIPW) estimator
>
> This is a very good and subtle question! We will add more discussion in the next version.
>
> To begin, we claim that we only borrow the idea of the PPI estimator from a semi-supervised perspective, while our idea is different from not only PPI but also AIPW, especially due to the complexity of network interference compared to classic (i.i.d. or MACR) scenarios. In summary, our idea is **debiasing using prediction** or prediction-powered debiasing, in comparison to the traditional **debiasing of the (pure) prediction**.
>
> First, the PPI estimator and the doubly robust estimator are important but different ones in semi-parametric inference. Our form embodies a **standard** point estimate of PPI, as presented in *Angelopoulos et al. (2023a)*, while, as discussed in *Angelopoulos et al. (2023b)*, in the task of mean estimation, the AIPW can be reorganized as a PPI++ estimator with **a particular choice of power tuner**. Moreover,  PPI estimator involves no nuisance estimator (AIPW involves two nuisance estimators) and behaves much differently from AIPW estimator in score space (as discussed in *Predictions as Surrogates: Revisiting Surrogate Outcomes in the Age of AI*).
>
> Second, concerning targets, the PPI-style estimator focuses more on the assumption-lean case and the prediction function $f$ is truly treated as an unknown black box, pursuing variance reduction as long as $f$ is correlated to $Y$. On the other hand, the AIPW estimator pursues consistency and semi-parametric efficiency when (at least one of) the two nuisances is **well-specified**.
>
> Finally, in our scenario, the data generation mechanism is much more complex beyond the classic i.i.d. data. For example, though we view the interior nodes as **labeled data**, the probability of a node becoming an interior node is usually too complicated to be calculated. We borrow the idea from PPI, while the story is actually different: even the sample mean of outcomes of interior nodes (MII estimator) can incur selection bias (mean of Y in labeled data is unbiased in classic i.i.d. setting), and we **use prediction to mitigate the bias stemming from systematic covariate discrepancy between interior and boundary units**. The rationale is that the regression function can model covariates flexibly while incurring the difficulty of extrapolation of the treatment proportion.

---

> > ### Author Response · Authors · 2025-11-13
> >
> > ## Assumption 4.1
> >
> > This assumption and Theorem 4.1 are both designed to **illustrate the behavior of the AMII estimator**. To enable an exact bias analysis, we require not only the potential outcome model (Equation 13) but also an explicit form of the regression component—hence its introduction.
> >
> >
> >
> > ## One dataset
> >
> > In fact, we also provide results on **another network topology** in the appendix (see Table 14/15 and Section A.4).
> >
> > Moreover, the **data generation mechanism**—rather than the network topology—is the most critical factor influencing performance. This is consistent with prior literature, where studies typically consider one or two networks and instead vary the potential outcome model. In our work, we further examine three resolution levels of the clustering algorithm to assess the impact of cluster structure (see Tables 2–7).
> >
> >
> >
> > ## Figure 3 issue
> >
> > We note that these statistics are computed on a **billion-scale social network**. In the absence of finite-sample concerns, such distributional discrepancies are considered highly significant by our industrial collaborators and are expected to correspond to **substantial selection bias**.
> >
> >
> >
> > ## Question 1
> >
> > Cluster-level randomization is already presumed in our setting (as stated in the title, abstract, and Line 165), so we believe it is unnecessary to restate it. Without this assumption, the entire framework involving cluster structure and interior nodes would not hold. This randomization scheme is also standard practice in large-scale industrial experiments. Consequently, it is typically **not formulated as an explicit assumption** in related literature, since it is part of the known experimental design.
> >
> > In addition, Assumption 4.1 and Equation (13) are explicitly referenced in the statement of Theorem 4.1.
> >
> >
> >
> > ## Question 2
> >
> > Assumption 4.1 specifies the form of the regression estimator, where $g$ denotes a learned transformation function within this estimator (e.g., a MLP). In contrast, Equation (13) represents the data generation mechanism assumed in Theorem 4.1. To clarify this distinction, we have already added hats in Equation (14) to indicate the estimated quantities.
> >
> > ## Question 3
> >
> > See **the Comparison with Doubly-robust (AIPW) estimator** discussed before.
> >
> > ## Question 4
> >
> > This means that when each $Y_i$ is an unbiased estimate of a common parameter $\theta = E[Y_i]$, the estimator $\hat\theta = \sum_{i=1}^m w_i Y_i$ achieves minimum variance among unbiased estimators when $w_i = 1/m, \forall i \in [m]$. Introducing unnecessary weights, such as the bi-level weights in CAE, increases the variance in this setting. Moreover, incorporating outcomes from boundary nodes could further reduce variance but at the cost of introducing bias. Due to space limitations, we omitted these details from the main paper.
> >
> >
> >
> > ## Question 5
> >
> > There is usually no single ground truth for clustering, as each graph clustering (or community detection) algorithm has its own inductive bias and produces a distinct cluster structure. We also systematically examine the influence of clustering resolution—which affects both the cluster size and the number of clusters—in Tables 2–7.
> >
> >
> >
> >
> >
> > ## **Minor issues**
> >
> > - We thank the reviewer for their careful reading. We are referring to **Table 10** in the Appendix and have added the corresponding reference in Line 466. When there is no multi-hop interference and no covariates, the case indeed reduces to that in Equation (11).
> > - The discrepancy is statistically significant (e.g., $\alpha = 0.05$) based on a two-sample K–S test.
> > - We will add punctuation marks where necessary in next revision.
> > - Yes, In Line 475, it should be *polynomial in* $p$, and we have corrected this.
> > - $c_i$ denotes the number of **distinct connected clusters** for unit $i$, while $\deg_i$ represents its **degree**, i.e., the number of directly connected units. A unit $i$ is said to be connected to cluster $k$ if it connects to at least one unit in that cluster.

---

### Official Review · Reviewer_gJGo · 2025-11-01

**Soundness:** 4
**Presentation:** 4
**Contribution:** 2
**Rating:** 6
**Confidence:** 5

**Summary:**

The paper proposes a new estimator for the global average treatment effect (also known as the total treatment effect) in the presence of network interference. The work falls under the framework of cluster randomization, where conventional methods (e.g. difference in means or Horvitz-Thompson) suffer from either bias due to inter-cluster interference or large variance due to randomization. Specifically, the authors propose to focus on the interior units whose potential neighbors are all within the same cluster in order to reduce the variance. Furthermore, the bias from sub-sampling is corrected (or partially corrected when mis-specified) by a counterfactual predictor. Under certain assumptions, the bias bound is carefully elaborated, and more simulations are added for model misspecification.

**Strengths:**

- **originality**: the proposed method focusing on interior units is new and novel.
- **quality**: the paper is well-written with solid theoretical results and simulations.
- **clarity**: the paper is clear and well-explained assumptions
- **significance**: the significance is relatively fair (see weakness part).

**Weaknesses:**

Although the paper provides a thorough analysis of the bias of the proposed method, the variance analysis part of the MII estimator is missing, which makes it hard to conduct a real test of the causal effects. More detailed questions are left to the questions part.

**Questions:**

1. How can we use the proposed method to do hypothesis testing on GATE? Is there any direct way that we can estimate the variance of the MII estimator along with the asymptotical normality property?
2. I'm a bit confused about the asymptotic scheme the assumption 3.1 is working on. Are we considering a sequence of networks with an incremental number of clusters? It could be more helpful if the authors can show that the assumption 3.1 is easy to satisfy for common random network models. The $o_p(K^{-1})$ bound for the maximum deviation does not sound trivial to me. My intuition for the bound is around $O_p(K^{-1/2}\log K)$ for Erdos-Renyi random networks or similar.
3. The bound in Theorem 3.1 does provide a convergence rate related to either cluster size or number of clusters. It could be refined to provide more insights on designing an experiment.

---

> ### Author Response · Authors · 2025-11-13
>
> We sincerely thank the reviewers for their recognition of our paper and for their efforts during the review process. We clarify some issues and settle the concerns below.
>
> ## Variance analysis
>
> First, as illustrated in Figure 2 and Tables 2–11, we empirically observe that the variance is much less prominent than the bias, since the primary issue introduced by network interference lies in the latter. The intensity and range of interference mainly affect the **bias**, while their influence on variance is relatively minor. Therefore, we focus on characterizing and mitigating bias.
>
> Second, analyzing **variance** is substantially more involved than analyzing bias, as it requires specifying both a concrete experimental design and, typically, a parametric potential outcome model (e.g., Equation (11) in our paper). Moreover, the most challenging aspect arises from cluster-level randomization, which induces treatment dependencies determined by the network topology and often renders exact analysis intractable. For instance, Theorem 1 in *Chen & Li (2024)* demonstrates that even for a simple linear regression–based estimator, the variance expression can be highly complex.
>
>
>
> ## Inference on GATE
>
> This is indeed an excellent point, and it aligns well with what our industrial collaborators are pursuing. Unfortunately, given a single experiment, valid inference for the GATE is only feasible under
>
> - a carefully designed experimental scheme,
> - a set of strong structural assumptions on the network topology.
>
> We take *Kandros et al. (2024)* as an illustrative example to highlight the challenges of conducting inference for GATE. As far as we know, this work provides the best theoretical results on the HT estimator and explicitly targets inference for the GATE (rather than surrogate estimands such as exposure effects). In their Theorem 5.6, a CLT is established under Assumptions 1–5 (together with the neighborhood interference assumption). A crucial condition therein is $d_{\max }(\mathcal{H})=o\left(n^{1 / 9}\right)$, where $\mathcal{H}$ denotes the *conflict graph*. Intuitively, as the estimand depends on treatments of an increasing number of units (i.e., approaches the GATE), the conflict graph becomes denser. Under the neighborhood interference assumption, when the estimand is the GATE, the conflict graph becomes $\mathcal{H} = \mathcal{G}^2$ (the two-hop connected graph), meaning that two nodes are adjacent in $\mathcal{H}$ whenever their graph distance in $\mathcal{G}$ satisfies $l(i,j) \le 2$.
>
> In practice—e.g., for billion-scale social networks—this condition imposes an unrealistic constraint on network density. Even in our running example (with over 10,000 units), applying such a design leads to fewer than 10 selected units per realization, making meaningful inference infeasible. Theoretically, satisfying this restrictive assumption would require a highly regular and sparse network structure, such as an $r$-regular graph with small constant degree $r$.
>
> If the estimand becomes more local—e.g., direct treatment effects or certain exposure effects rather than the GATE—the corresponding assumptions can be relaxed, though the intrinsic difficulty remains substantial.
>
> Returning to our setting, even under strong structural assumptions, the presence of **cluster-level randomization**—especially when clusters are algorithmically generated—renders variance analysis nearly intractable. Although a formal variance expression can sometimes be derived (e.g., Theorem 4.2 in *Liu et al., 2024*), it typically offers little insight, as it involves numerous unidentifiable or unestimable nuisance components.

---

> ### Author Response · Authors · 2025-11-13
>
> ## Asymptotic regime
>
> This is indeed a very interesting point. First, we clarify that our analysis considers a sequence of networks with both the number of units ($n$) and the number of clusters ($K$) increasing simultaneously. We exclude the case where $n$ increases while $K$ remains fixed.
>
> Although Assumption 3.1.1 serves primarily as an auxiliary bridging assumption, it nevertheless imposes a nontrivial restriction on the asymptotic regime. Even under a specific network generation mechanism (e.g., an Erdős–Rényi graph), the corresponding rate critically depends on both the network topology parameters (such as the edge probability $p$) and the cluster generation mechanism. For example, the Louvain/Leiden algorithms often produce clusters with highly variable sizes, while algorithms such as METIS tend to generate more balanced partitions. Consequently, the convergence behavior of the representativeness term can vary in a complicated and nuanced manner.
>
> To illustrate, consider an Erdős–Rényi graph with $p = 5 \times 10^{-4}$ and $n = 10^4$ (yielding an average degree of approximately $5$). Using the Louvain algorithm with its default resolution level ($\gamma = 1$), we obtain $K = 112$ clusters and 1,129 interior nodes. The representativeness discrepancy measure is
> $$
> \max_{i \in[K]}\left|\frac{\left| Int_i\right|}{\sum_{k \in[K]}\left| Int_k\right|}-\frac{n_i}{\sum_{k \in[K]} n_k}\right| \approx 0.005
> $$
> In this case, the result suggests we are more likely in the regime $o_p(1/K)$ rather than $O_p(\log K / \sqrt{K})$. However, by adjusting parameters such as $p$ or $\gamma$, we may shift to another regime, which reinforces our view that this assumption imposes a substantial restriction on the overall data generation mechanism.
>
> In addition, we note that the Erdős–Rényi model itself may not fully capture the characteristics of real-world social networks, where the average degree typically grows very slowly with network size. For example, in industrial-scale networks with billions of nodes, the average degree often remains around 1,000—comparable to that of networks several orders of magnitude smaller.
>
> Finally, we provide some technical discussion. We need the Assumption 3.1.1 to prove:
> $$
> \frac{\sum_{k\in [K]} m_k \bar Y_{k, Int}(\mathbf{1})}{\sum_{k\in [K]} m_k}
>     -
>     \frac{\sum_{k\in [K]} m_k \bar Y_{k, Int}(\mathbf{0})}{\sum_{k\in [K]} m_k}
> =
> \frac{\sum_{k\in [K]} n_k \bar Y_{k,Int}(\mathbf{1})}{\sum_{k\in [K]}  n_k}
>         -
>         \frac{\sum_{k\in [K]} n_k \bar Y_{k,Int}(\mathbf{0})}{\sum_{k\in [K]} n_k}
>         +o_p(1)
> $$
> Hence, we choose to restrict the gap between two positve sequences $m_k/\sum_{i=1}^K m_i$ and $n_k/\sum_{i=1}^K n_i$. A weaker alternative is to assume that the $\ell_1$ distance between these two sequences is $o_p(1)$, together with bounded mean outcomes.
>
> In contrast, *Liu et al. (2024)* adopt a stronger assumption on the outcomes. Specifically, by assuming that the expectations of the mean outcomes under global treatment and control are identical constants (the “common mean” assumption; see Assumption 4.3.1 in *Liu et al.*, 2024), the restriction on these two proportion sequences can be substantially relaxed. However, such a common-mean assumption is evidently much more restrictive in practice.
>
> (Additionally, we have corrected a minor typo in the denominator summation in Equation (22) and (23): the summation index should be $\sum_{k \in [K]}$ rather than $\sum_{j \in Int}$.)
>
>
>
> ## Enlighting experimental design
>
> We clarify that Theorem 3.1 only establishes a convergence result (consistency) rather than a convergence rate. On the one hand, our paper primarily focuses on estimation techniques rather than the optimization of experimental design.
>
> On the other hand, the experimental design itself can indeed be optimized, as its key **design parameter** can be incorporated into the objective (e.g., the mean-squared error of the estimator). For example, *Leung (2022)* optimizes the number of clusters in a grid setting to achieve rate-optimal MSE with cluster-randomized designs under spatial interference (*Michael P Leung. Rate-optimal cluster-randomized designs for spatial interference. The Annals of Statistics, 50(5):3064–3087, 2022.*). Moreover, *Chen et al. (2023)* optimize the covariance matrix of the treatment vector. The former considers a grid structure (simpler than a general network) and derives the optimal rate of the MSE to determine the appropriate number of clusters, while the latter relies on a parametric potential outcome model and standard HT estimator without network exposure.
>
> The position of our paper is estimation techniques, and we do not have such a design parameter that can be injected to the MSE.

---

### Official Review · Reviewer_JmRm · 2025-11-01

**Soundness:** 2
**Presentation:** 3
**Contribution:** 2
**Rating:** 4
**Confidence:** 4

**Summary:**

The paper addresses the challenge of estimating the Global Average Treatment Effect (GATE) in the presence of network interference, a scenario where a unit’s outcome is influenced not only by its own treatment but also by the treatments of its neighbors. The authors focus on cluster-level randomization and identify limitations in existing estimators and propose new mean-in-interior estimator and augmented mean-in-interior to address the limitations in existing estimators for estimating GATE. Simulation studies are conducted to assess and validate the effectiveness of the proposed estimators and compare with existing estimators across a range of network and treatment settings.

**Strengths:**

1.  The paper addresses the important problem of estimating the average treatment effect under network interference, which has practical significance for causal inference in many real-world applications.

2. The motivation to extend existing estimands is well-articulated, and the comparisons with prior work are thoroughly examined.
3. Viewing interior nodes as "biased labeled data" and approaching the problem from a semi-supervised learning angle is conceptually compelling.
4. The core idea is clearly explained and easy to follow.

**Weaknesses:**

1. The proposed estimators rely on strong parametric assumptions, which may limit their robustness and applicability in real-world settings. Moreover, the practical implementation of these estimators could be challenging, particularly in large-scale or heterogeneous networks where model assumptions may not hold or be verifiable.
2. The impact of network interference on the proposed estimators is not thoroughly addressed. In particular, the paper does not fully analyze how varying levels or structures of interference may affect estimator performance, which raises concerns about their robustness in complex or sparse networks.

**Questions:**

1. The nodes within each cluster are partitioned into interior and boundary nodes. In sparse networks, the number of interior nodes may be relatively small, which could adversely impact the performance and stability of the proposed estimators. Could the authors elaborate on how the estimators behave under such conditions, and whether any adjustments are needed to ensure reliable performance in sparse or low-density network settings?
2. The paper proposes mitigating selection bias in the Mean-in-Interior estimator by training a counterfactual predictor over the entire graph. Could the authors elaborate on the rationale behind this approach?
3. In the proposed approach, boundary nodes are excluded and only interior nodes are retained for estimating the global average treatment effect under network interference. However, interference is inherently present in GATE since it captures both direct and indirect effects propagated through the network. How do the authors account for or mitigate the influence of network interference on the estimation process, given that boundary nodes—where interference effects may be most pronounced—are omitted from the analysis?

---

> ### Author Response · Authors · 2025-11-13
>
> First, we sincerely thank the reviewer for the efforts in reviewing our paper. We clarify some issues and settle the concerns below.
>
> ## Robustness
>
> On the one hand, for theoretical analysis of causal inference under network interference, parametric assumptions are often inevitable—particularly when the estimand becomes more global (i.e., involving the treatment assignments of more units). For instance, such assumptions are also employed in the analysis of CAE and Hajek estimators in our references. We also clarify that some parametric assumptions, such as the potential outcome model in Equation (11), are used primarily for **illustration**. In contrast, Assumptions 2.1 and 3.1 are high-level conditions that support Theorem 3.1 without specifying any particular model form.
>
> In fact, given the difficulty of GATE estimation, most technical assumptions in the existing literature are difficult to verify in practice (see, for example, *Leung & Loupos, 2024*). Nonetheless, the core component in Assumption 3.1—the **representativeness** assumption—can be approximately validated using historical experiments of similar treatment types, as confirmed by our industrial collaborators. For example, when repeatedly testing different sizes of a core user interface component, the representativeness of interior nodes across these experiments is often stable. Such verification can help determine whether incorporating a regression component is necessary.
>
> On the other hand, we emphasize that the computation of both the MII and AMII estimators does **not** require estimating any parameters within parametric models; they rely solely on observed treatments, outcomes, and the given network topology.
>
> Finally, in our simulation study (Tables 2–11), we systematically examine the robustness of both estimators under model misspecification—for instance, under two-hop interference (violating Assumption 2.1) and unrepresentative interior nodes (violating Assumption 3.1).
>
> ## Network impact
>
> **Exact** analysis of the bias and variance of even a simple estimator for GATE estimation under network interference is highly challenging and inevitably requires a parametric potential outcome model (see, for example, *Yu et al. (2022a)*). In our paper, under the potential outcome model in Equation (11), we show that our estimator is unbiased, whereas its variance is highly complex, involving numerous cluster-level statistics of the network topology—particularly under cluster-level randomization.
>
> Moreover, as illustrated empirically in Figure 2 and Tables 2–11, the intensity and range of interference primarily affect the **bias**, while their influence on variance is relatively minor. Therefore, our theoretical analysis and illustrations focus on characterizing and mitigating bias.
>
>
>
> ## Sparse network
>
> In summary, as the network becomes sparser, the impact of network interference also diminishes, making the problem considerably easier. For example, *Chen et al. (2023)* show that the bias of the standard HT estimator converges to zero as the number of between-cluster edges decreases, and a similar trend holds for the variance.
>
> In addition, the number of interior nodes will not converge to zero, since graph clustering is typically adaptive. At the very least, the clustering resolution can be adjusted according to network sparsity, ensuring that our methods do not degrade. When the network becomes extremely sparse—comprising many disconnected components—the setting naturally reduces to the “partial interference” regime, where all nodes are interior nodes. In that case, simple cluster-level randomization has already been shown to be effective.
>
> To illustrate, we set $p=5\times 10^{-4}$ and $n=10^4$ for an Erdős–Rényi graph ($\overline \deg = 5$). Using the default resolution of the Louvain algorithm ($\gamma=1$) induces $K=112$ clusters and 1129 interior nodes. This configuration yields a similar number of nodes to the Facebook network in our running example but is substantially sparser (in terms of average degree). Nevertheless, the proportion of interior nodes **increases** to 11%, compared with 8% in our case.
>
>
>
> ## Rationale of mitigating bias
>
> In our paper, Theorem 4.1 and Section 4.3 are designed to provide intuition for the bias reduction achieved by our adjustment procedures. In brief, the source of selection bias lies in the systematic differences between the subpopulations of interior and boundary units, which are characterized by their covariates. Traditional statistical estimators without regression components cannot address this issue, as they rely heavily on interior units (including our MII estimator). In contrast, the regression-based adjustment leverages a prediction model for counterfactual outcomes that captures both the covariates and their interactions with treatments, thereby mitigating the bias arising from covariate discrepancies between the two subpopulations.

---

> ### Author Response · Authors · 2025-11-13
>
> ## Question 3
>
> First, we clarify that it is a common misconception that the interference effect is more pronounced for boundary nodes. For example, consider the classic linear-in-means model:
> $$
> Y_i(\mathbf{z})=\beta_0+\beta_1 z_i+\frac{\sum_{j \in \mathcal{N}(i)} z_j}{\operatorname{deg}_i}
> $$
> In this setting, the interference effect of a treated **interior** node is the most prominent (in magnitude) among all units—specifically, it equals 1. In contrast, for **boundary** nodes, the interference effect (i.e., the treated proportion of 1-hop neighbors) typically concentrates around the overall treatment proportion $p$, which is usually much smaller than 1 in practice.
>
> Second, estimating GATE does **not** necessitate interpolating between all counterfactuals from global control to global treatment. In fact, it suffices to accurately estimate only these two extreme counterfactuals.
>
> Next, as discussed in the Introduction (particularly Figure 1 and the accompanying explanation), existing network-aware estimators—though not always explicitly described as such—substantially **exclude most boundary units**, especially when the treatment proportion $p$ is small.
>
> For illustration, consider the case where $p=0.1$ (a typical intermediate treatment proportion in practice). In expectation (via Monte Carlo repetition), there are about 102 fully treated nodes, of which only 7 are boundary nodes. The HT, CAE, and Hájek estimators only retain these fully treated nodes. Although interior nodes constitute merely 8% of all nodes, they account for the vast majority of units remaining after this trimming. Therefore, this issue is not unique to our MII estimator but inherent to all such estimators. Our estimator leverages this small fraction of boundary units at the cost of additional weighting on outcomes, which naturally increases variance. Consequently, the MII estimator consistently achieves the lowest variance among the five methods, as demonstrated in our simulation results.
>
> Regarding the theoretical analysis, Assumption 3.1 states that the outcomes under global treatment for interior nodes are representative of those for the entire cluster, which includes both interior and boundary nodes. Hence, we do not omit boundary nodes in our analysis.
>
> Finally, we note that counterfactual prediction in the AMII estimator is performed on **all** units, naturally including boundary nodes. We also provide an ablation study where the regression is restricted to boundary units only (see Tables 12 and 13). This demonstrates how boundary nodes participate in the estimation process of AMII estimator.

---

### Author Response · Authors · 2025-11-14

We have synthesized the suggestions and questions proposed by reviewers and updated a new version of the paper (PDF). Specifically, we mainly:
- add connecting and explanatory sentences for enhancing the flow and clarifying the position of our paper,
- add more intermediate steps and explanations in our proof
- fix typos

---

### Author Response · Authors · 2025-11-28

# Position Summary

This paper targets estimating the global average treatment effect under network interference. Network interference has emerged as a critical issue in modern platforms such as social networks and two-sided markets. It introduces significant bias and undermines the accuracy of A/B testing, which is widely regarded as the gold standard for product launch and iteration in industry. In academia, network interference is also a prominent research topic, particularly because it poses both intrinsic and practical challenges, giving rise to a range of new problems as it violates the classic Stable Unit Treatment Value Assumption (SUTVA).

Based on extensive numerical simulation and observations, we recognize the over-reliance of existing statistical estimators on the interior nodes of clusters, and we first propose a Mean-in-Interior (MII) estimator that **removes inefficiency brought by unnecessary weights** on potential outcomes, which achieves substantial variance reduction. We further recognize the potential **systematic bias due to the discrepancy between interior and boundary units** (we also provide a desensitized empirical evidence from a realistic large-scale  social network in industry for this observation), and propose the adjusted MII estimator that embodies a form of point estimator in prediction-powered inference, as well as a semi-supervision perspective. In general, our novel adjustment can be viewed as **"debiasing using prediction"**, which we further elaborate in our responses to reviewers.


To validate the efficiency of our estimators, we construct the consistency of the MII estimator based on much weaker assumptions in the literature, and the bias analysis of the AMII estimator to illustrate the bias reduction. We further conduct a**comprehensive simulation study with challenging settings** (violation of representativeness assumption, low signal-to-noise ratio, multi-hop interference, etc.) and abundant ablation studies to demonstrate the advantages of our two estimators.

# **Rebuttal Summary**

Upon receiving the feedback of reviewers, we realize that there are many misunderstandings regarding our paper. On Nov 13, we uploaded a revised manuscript to improve clarity and provided detailed responses to every weakness and question raised by the reviewers. Prior to reverting, we received feedback from Reviewer RLAt, who raised their score to 6 (with confidence adjusted to 3). However, due to the freeze of rebuttal, we regretfully can no longer discuss with the other three of our reviewers. Hence, besides the position summary to clarify the position and contribution of this paper, we also supplement this rebuttal summary (also for the convenience of AC), and we believe that our responses are sufficient to address the points proposed by our reviewers.

We summarize the primary concerns and our brief responses below. Detailed rebuttals and responses to specific questions can be found in the official comments for each reviewer.


- **Comparison with cluster-adaptive estimator (CAE)**: In the MII estimator, we remove the unnecessary weights on the potential outcomes and improve the efficiency consistently. In the AMII estimator, we solve the common (but not handled) problem in both the CAE and MII estimator: over-reliance on interior nodes, which is widely existing due to the heuristic nature of clustering and would introduce systematic bias.

- **Comparison with AIPW (doubly-robust) estimator**: Our AMII estimator embodies a PPI point estimator and targets "debiasing (pseudo-)labeled data using predictions" in the scenario of complex network interference, while both the PPI point estimator and AIPW estimator target "debiasing the predictions using labeled data".

- **Variance analysis**: There are three-fold complexities hindering a meaningful variance characterization: regression model, network interference (outcomes interdependency), and cluster-level randomization (network-based treatments interdependency). We also illustrate the intrinsic difficulty of inference with GATE (a global estimand) as shown in *Kandros et al. (2024)* and the complexity of variance of the estimator with regression-component as shown in *Chen et al. (2024)*. In addition, we supplement the discussion on the efficiency gain of the MII estimator.

- **Thin experiments**: Due to the nine-page constraint, we leave most of the experiments to Appendix A, and wemention them in our main paper (with corresponding \ref), and we guess some of our reviewers may miss. In fact, we systematically examine our two estimators under comprehensive and challenging settings with necessary ablation studies.

---

### Meta-Review · Area_Chair_Xao1 · 2026-01-06

**Summary:**

The paper proposes a new estimator ("MII") to estimate treatment effects in the presence of network interference. The goal is to simultaneously improve the bias of prior estimators, and improve the robustness by not requiring reweightings (which also helps with variance). An augmented variant of MII with a trained counterfactual ("AMII") is also introduced, and both show good empirical performance across a variety of ablation studies. Reviews were initially mixed, but all reviews highlighted the relevance of the problem formulation, the conceptual elegance of the results, and the writing quality. I believe the reviewers did a good job of rebutting to the various points and clarifying misconceptions, and I am optimistic towards the value of this work. I recommend acceptance.

That said, I do believe the discussion with the reviewers added clarity to the paper's contributions, and request that the authors accordingly merge this discussion into the paper in the final version.

**Reviewer Concerns:**

The reviewers provided mixed opinions, but I concur with the authors that there seems to be some misunderstandings re: assumptions, breadth and convention of experiments, and technical novelty, that are adequately addressed by the rebuttal.

**Reviewer Scores:**

The reviewers averaged a score of 4 before rebuttals. One of the reviewers explicitly mentions raising their score, and the authors mention it was raised to a 6 (which seems consistent with the discussion thread), which would mean an average score of 5. Also, even the relatively weaker reviews had significant strengths mentioned, so I think it's plausible these scores would also be raised.

---

### Decision · Program_Chairs · 2026-01-26

Accept (Poster)